# Unprecedented UK heatwave harmonised drivers of fuel moisture creating extreme temperate wildfire risk
Katy Ivison [1] ✉, Kerryn Little[1], Alice Orpin[1], Claire M. Belcher[2], Gareth D. Clay[3], Stefan H. Doerr [4], Thomas E. L. Smith[5], Roxane Andersen [6], Laura J. Graham[1,7] & Nicholas Kettridge [1]

Climate change is resulting in more extreme fire weather during major heatwaves. Across temperate Europe, shrub landscapes dominate the area burned, with the moisture content of fuels during these events determining the threat posed. Current controls on the moisture content of temperate fuel constituents and their response to future extreme heatwaves are not known. We took field measurements of live and dead heather (*Calluna vulgaris*) and organic soil moisture content across the UK over 3 years, including an intensive sampling campaign during the July 2022 heatwave. Here, we show that the fuel moisture content of live fuel is associated significantly with phenological variables, dead fuel only with weather variables, whilst organic-rich ground fuels are more associated with landscape variables. However, during the record 2022 heatwave there was a harmonisation in fuel moisture controls across different fuel constituents, with those controls being driven by weather alone. This caused synchronised extreme dryness outside of current seasonal norms across all fuel constituents at the same time and place. Future intense summer heatwaves can therefore be expected to align the most severe conditions for fire ignition, spread and impact in traditionally non-fire prone regions, producing humid temperate landscapes susceptible to extreme wildfire events.

In the last two decades, Europe has witnessed a growing prevalence of extreme heatwaves[1]. These prolonged periods of exceptionally high temperatures, persisting for days to weeks, are attributed to the influence of blocking high-pressure systems[2]. In July 2022, the UK experienced an unprecedented extreme heatwave as a result of a 'heat dome', where high pressure pushes warm air downwards and traps it at the surface[3]. Temperatures exceeded 40 °C for the first time and the UK Met Office issued a red warning for extreme heat[3]. In fact, this event was ranked as the highest intensity July heatwave since records began in 1878, based on mean and maximum temperatures, and broke temperature records across the country[4] (Fig. 1a). Human-induced climate change is anticipated to intensify heatwaves[5]. Three heatwave events have occurred in the UK within 2025 alone and Europe has experienced four record-breaking heatwaves since 2003[1]. It is therefore likely that such events will become more frequent and severe in the future—in the UK, the number of heatwave days are predicted to increase by up to 2 days every decade[6]. Marked by elevated temperatures

and reduced humidity, heatwaves create conditions conducive to extreme fire weather[7,8] and the drying of fuels, leading to lower moisture contents and a high risk of both wildfire ignitions and extreme fire behaviour. The UK July 2022 heatwave coincided with an unprecedented number of wildfires, overwhelming Fire and Rescue Services and causing damage to many properties[9].

The drying response of fuels to extreme weather in traditionally fire-prone Mediterranean and continental climates is established[10,11]. Within these climates, fuel moisture and associated fire danger are determined predominantly by fire weather[12,13]. However, we lack understanding of fuel moisture dynamics in traditionally non-fire-prone humid temperate regions. As a result, these regions are now grappling with a surge of high-intensity wildfires and extreme fire behaviour without the fundamental knowledge of fuel moisture dynamics, both now and under future extremes[14]. This presents a significant challenge to firefighting services and land managers, leaving them ill-equipped to confront heightened

[1]School of Geography, Earth and Environmental Sciences, University of Birmingham, Birmingham, UK. [2]wildFIRE Lab, Hatherly Laboratories, University of Exeter, Exeter, UK. [3]Department of Geography, School of Environment, Education and Development, University of Manchester, Manchester, UK. [4]Department of Geography, Centre for Wildfire Research, Swansea University, Swansea, UK. [5]Department of Geography and Environment, London School of Economics and Political Science, London, UK. [6]Environmental Research Institute, University of the Highlands and Islands, Thurso, UK. [7]Biodiversity, Ecology and Conservation Group, International Institute for Applied Systems Analysis, Vienna, Austria. ✉e-mail: k.e.ivison@bham.ac.uk

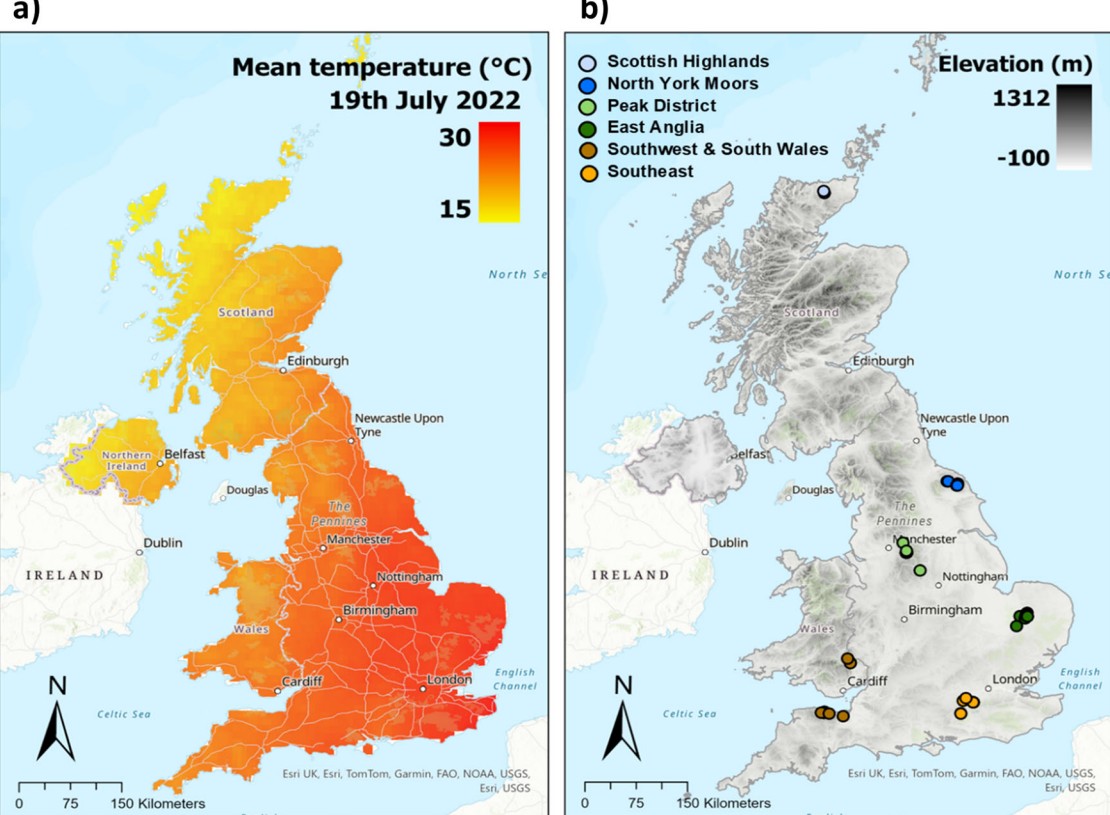

**Fig. 1 | UK heatwave temperatures and study sample sites. a** Mean temperature of main heatwave day (19th July 2022) across the UK, (**b**) distribution of sample sites and their associated climate regions. Map sources: Esri, DeLorme, HERE, TomTom, Intermap, increment P Corp., GEBCO, USGS, FAO, NPS, NRCAN, GeoBase, IGN, Kadaster NL, Ordnance Survey, Esri Japan, METI, Esri China (Hong Kong), swisstopo, MapmyIndia and the GIS User Community. Temperature data is from the E-OBS ensemble gridded dataset version 26.0[57].

threats without the knowledge to develop effective landscape management strategies to mitigate the escalating risks posed.

In humid temperate climates such as in the United Kingdom, heathlands and peatlands dominate burned areas, with a substantially higher (4–7×) number of fire detections per unit area compared to forests or arable land covers (with the greatest burnt area in the UK occurring in mountain, bog and heath[15]). In the UK, such heathlands and peatlands are often characterised by the dwarf shrub *Calluna vulgaris*, a fire-adapted species which regenerates after burning[16], though regeneration is poorer within older *Calluna* plants[17]. Peat is a particularly good substrate for *Calluna* seedling growth following fire[18]. Many UK wildfire studies have focused on *Calluna*-dominated heathlands and peatlands (e.g. Davies et al.[17], Davies and Legg[19], Davies et al.[20]).

Live *Calluna* provides the dominant fuel for fire spread[21]. *Calluna* fuel moisture exhibits complex dynamics[22], with ecohydrological drivers differing from the weather-dependent controls of dead fuels that underpin fuel moisture models in traditionally fire-prone regions (e.g. Van Wagner & Pickett[23]). Notably, phenology and landscape factors provide strong impacts on the fuel moisture content of live vegetation[24,25]. The moisture of live *Calluna* is lowest in the spring before the summer green up[22,26]. As a result, large wildfires occur more frequently during spring than in warmer, drier summer conditions[27,28]. The moisture content of dead *Calluna*, which consists of either dead branches of live plants or dead plants which have not yet decomposed, is controlled by environmental conditions[27] and affects probability of fire ignition[19]. Carbon-rich ground fuels also provide an important fuel source for humid temperate wildfires. High-severity fires trigger high carbon emitting smouldering ground fires in organic soils when the moisture content of moss, litter and soils is low[20,21]. Landscape factors, such as soil type and topography[29,30], influence water retention and drainage

and in turn can regulate the moisture content of organic ground fuels[31] (Fig. 2).

We determined the relationship between the fuel moisture of live *Calluna* canopy, dead *Calluna* canopy and organic ground fuel, and a variety of weather, phenological and landscape variables. We then investigated how they, and the associated fire behaviour, respond to extreme heatwave conditions. 5845 fuel moisture samples were collected from 43 sampling sites (Fig. 1b) across five different climate regions of the UK, from the Scottish Highlands (58.4 °N) to the Southwest of England (51.1 °N), over a period of 3 years (2021–2023). We took advantage of the heatwave event of July 2022 by undertaking intensive field-based sampling across the sampling site network during the most extreme 3 days of the heatwave (107 samples in total), and explored how extreme heat impacts these drivers of fuel moisture. With these conditions expected to be more frequent by the middle of the 21st century[32], we simulated the fire behaviour under extreme heatwave and summer conditions and considered the implications of this future humid temperate wildfire behaviour on the UK Fire and Rescue services.

## Results

### Relationship between fuel moisture content environmental variables

Here, we use 'phenological' to refer to variables that represent seasonal shifts, comprising NDVI and long-term (20-year) mean daily temperature. 'Weather' refers to meteorological conditions such as rainfall and Vapour Pressure Deficit (VPD). Finally, 'landscape' refers to physical characteristics of a sample site, such as elevation, aspect and soil type. These results present a two-stage modelling approach. The first model (mixed effects model) estimates the effect of weather and phenological variables on fuel moisture content (FMC), using the sample site as a random effect. The second model

**Fig. 2 | Comparison of FMC seasonal trends across fuel types.** Seasonal variation of fuel moisture content (FMC) of live *Calluna*, dead *Calluna* and the organic soil layer. Live FMC is lowest in spring and increases in summer as green leaves emerge. Dead FMC is driven by short-term weather, increasing in wet conditions and decreasing in warm, dry conditions. Organic FMC is influenced by water retention and soil drainage so is at least partly driven by weather conditions, but this effect may be delayed. These drivers may shift under extreme weather conditions.

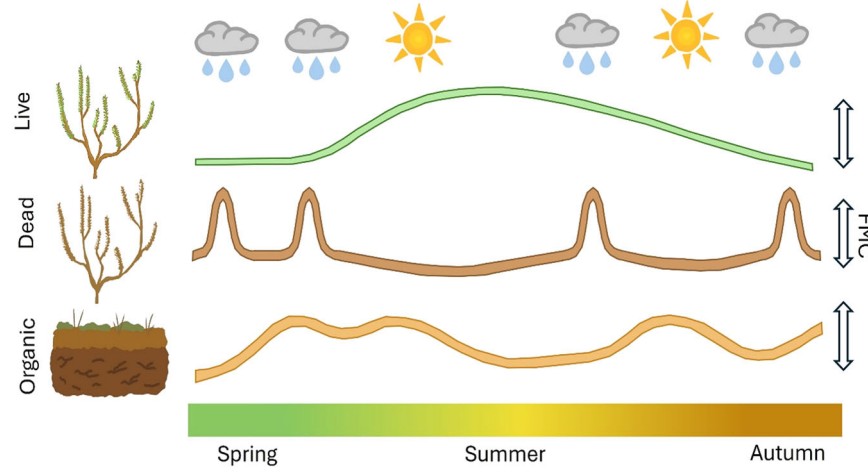

(linear model) takes as a response variable the site level random effect to estimate the effect of landscape variables on FMC.

Dead *Calluna* FMC was significantly associated with weather variables only (VPD and total rainfall of the sampling day and 5 days prior (hereon called five-day rainfall); $p < 0.01$ for both) (Fig. 3a). The model showed no significant influence of phenology on dead *Calluna* FMC. Weather and phenological variables explained 47% of dead *Calluna* FMC variability, but the site-level random effect explained 0% of FMC variability, indicating that landscape plays no significant role in dead *Calluna* FMC. Due to this lack of site-level variability, we did not run the landscape model for this fuel type. For live *Calluna*, in addition to the impact of VPD, FMC was also associated with phenological variables more than both dead *Calluna* and the organic layer; both long-term mean daily temperature and NDVI were significantly associated with FMC ($p < 0.01$ and $p = 0.02$, respectively) (Fig. 3b). Weather and phenological variables explained 23% of live *Calluna* variability, with a further 9% explained by site-level effects. Of these site-level effects, 37% of FMC variability was explained by landscape variables (Fig. 3c). The organic layer was significantly associated with a combination of weather variables and one phenological variable (long-term mean daily temperature) (Fig. 3d), but only 13% of FMC variability was explained by weather and phenological variables. 56% of FMC variability was explained by site-level effects, and of this, 61% of site-level variability was explained by landscape variables; both soil type and elevation were significant (Fig. 3e; $p < 0.05$). Organic layer FMC was therefore influenced by landscape to a much greater degree than the other fuel types.

## Heatwave impacts on fuel moisture—an analogue for future extremes

Intensive sampling took place between the 17th and 19th July 2022, during the most extreme conditions of the heatwave. At times of sampling, air temperature ranged between 24 °C and 42 °C and relative humidity ranged between 16% and 56% (Supplementary Table 1). The impact of the heatwave on live *Calluna* canopy FMC varied between regions. The live *Calluna* moisture of East Anglia and Southeast regions (average of 84% and 79% respectively; Fig. 4a; Supplementary Table 2) were substantially lower than during July 2021 (128%; FMC is calculated as the mass of water per mass of dry sample and can therefore exceed 100%) and were more comparable to the spring pre green up moisture contents in 2021 (74%) and 2022 (69%). Live *Calluna* moisture in the North York Moors and the Southwest were within the range of moisture contents observed during July 2021 (Fig. 4a). Dead *Calluna* canopy moisture during the heatwave was significantly below anything measured during spring 2021, July 2021 or spring 2022 (Wilcoxon tests, $p < 0.01$, Supplementary Table 3). Fuel moisture content in the heatwave averaged 4.2%, ranging from 3.2% to 6.0%, compared with an average of 17.7% in July 2021 (Fig. 4b; Supplementary Table 2). The organic

moisture content was also the lowest observed over the 3-year measurement period in each region (Supplementary Table 3; $p < 0.01$) and FMC was particularly low for East Anglia and the Southeast (5% and 27% respectively; Fig. 4c).

Heatwave fuel conditions produced a simulated surface fire rate of spread more than double that of spring and more than four times faster than typical July fuel conditions (Supplementary Table 4). Similarly flame length was approximately doubled when compared to spring and regular July conditions. Critically, the heat wave fuel conditions pushed the flame length to exceed 1.5 m which is the limit above which direct attack in firefighting using hand tools is not permitted[33]. Under heat wave conditions the probability of ignition rose to 87% with mean values for both spring and non-heat wave July <11%.

## Discussion

The differing seasonality of controls on the moisture content of the fuels associated with fire ignition (dead fuel moisture), fire spread (live fuel moisture) and smouldering combustion of ground fuels (organic fuel moisture)[19,21] within these humid temperate ecosystems likely supports a considerable resistance to wildfire under current climate conditions at the ecosystem scale (e.g. Anderson et al.[34]). Within peatlands and heathlands, during periods of high fire weather, notably during the summer, dead fuels are dry and fire ignitions are more probable[19]. However, typically during this period, the moisture content of the live canopy is high[27] (Fig. 4a) because of the important control of phenology on live FMC[35], despite the contrasting effect of VPD. In these ecosystems, the live *Calluna* canopy provides the dominant above ground fuel load and its typically high moisture content limits the potential for fire spread[21]. Further, modelled flame lengths remain low (Supplementary Table 4). The typically high live summer moisture contents will not support extreme fire behaviour in heathlands and peatlands[27] which contrasts with the summer fire regime of adjacent arable and grasslands ecosystems where fires occur predominantly under periods of extreme fire weather (e.g. Mozny et al.[36]). This difference in the seasonal timing of flammability conditions between ecosystems, driven by the phenological control of live *Calluna* moisture, reduces connectivity of fuels across landscape and limits the potential for large-scale fire spread. In comparison, in the spring, during periods of low live *Calluna* moisture, extreme fire weather conditions that dry the dead fuels and increase ignition probability are less likely and the fire danger in the adjacent landscape remains low. Finally, the moisture content of the organic ground fuels is strongly linked to soil type and elevation (Fig. 3c). Regions with the largest carbon stocks, such as the peaty soils at higher elevations (e.g. North York Moors, Peak District, South Wales/South West), retain the highest moisture contents (Fig. 4c; Supplementary Table 1), which limits the potential for smouldering combustion, have low sensitivities to the seasonal or short-term weather conditions.

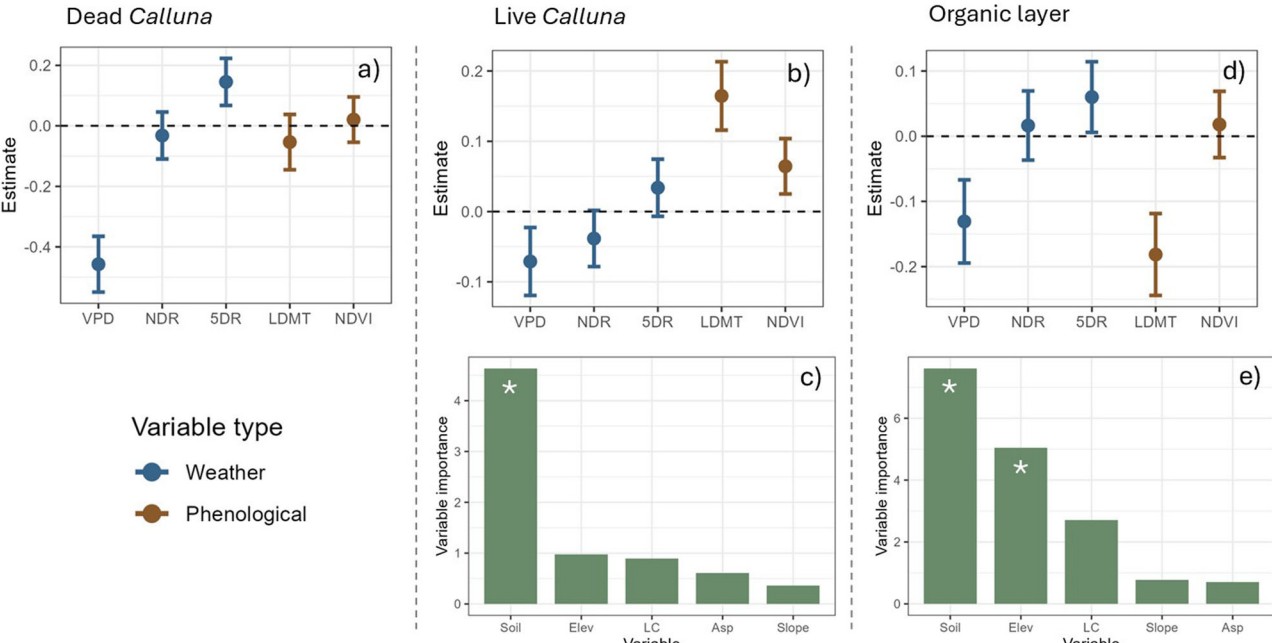

**Fig. 3 | Relationship between FMC and weather, phenological and landscape variables.** Two-stage models investigating which variables are associated with FMC for dead *Calluna* (**a**), live *Calluna* (**b**, **c**) and the organic layer (**d**, **e**). Forest plots (**a**, **b**, **d**) depict estimate (±95% confidence interval (CI)) of weather and phenological variables in first model: vapour pressure deficit (VPD), no. days since rain (NDR), five-day rainfall (5DR), long-term mean daily temperature (LDMT) and NDVI. Variables are significant if CI of estimate does not cross 0: negative estimate shows negative linear relationship between variable and FMC; positive estimate shows positive linear relationship between variable and FMC. Bar plots (**c**, **e**) depict the importance of each landscape variable (Elev: elevation, LC: land cover, Soil: soil classification, Asp: aspect) in second models, represented by *t*-value; the *t*-values for groups within categorical landscape variables (Soil, LC) were summed. Site variation for dead *Calluna* was close to 0 so landscape model was not carried out. Asterisks depict significant landscape variables (*p* < 0.05).

However, our results suggest that these seasonal controls on fuel moisture break down during extreme weather events. During the 2022 heatwave the moisture content of the dead *Calluna* canopy was extremely low. This low moisture content provides a highly ignitable fuel load[19], evidenced by the 87% probability of ignition (Supplementary Table 4). Whilst extreme, this was not surprising as our analyses (and studies within traditionally fire prone regions[11,23]) highlight the importance of only short-term weather variables, as opposed to phenological or landscape variables, on dead *Calluna* moisture. Live fuel moisture responses to extremes are far less well understood[37]. The change in moisture content of live *Calluna* and the organic layer during the heatwave was unexpected and has significant implications for the future fire proneness of this ecosystem.

We show that the phenological cycle for live *Calluna* canopy was strongly disrupted by the extreme weather of July 2022. This disruption was severe in East Anglia and the Southeast of England, particularly in lowland heaths. These regions are also where the highest temperatures and lowest humidity readings were observed during the heatwave (temperature range of 33 °C–43 °C and 33 °C–40 °C, humidity range of 16–26% and 19–39% for East Anglia and the Southeast respectively; Supplementary Table 1). The Southeast of England is where the greatest increase in heatwave events is predicted to occur[6], making this result even more significant in terms of future risk.

Similarly to live *Calluna*, we found that the FMC of the organic layer was lower during the heatwave per region than any other time period (Fig. 4c; Supplementary Table 3). This is of concern because for wildfires carried in surface vegetation, part of the heat is transferred to the organic layers beneath (e.g. duff, peat) and may ignite a smouldering fire[38] with significant carbon[20,39] and human health consequences[40]. During the 2022 heatwave, we observed that the organic layer became significantly drier than normal for the season in East Anglia and the Southeast, declining by up to two-thirds of their July average to just 12% and 38%, respectively. The organic layer moisture also declined to 85% in more northern regions where *Calluna* is typically underlain by carbon-rich peat deposits (Fig. 4c). Organic

root mat soils have been shown to have a 61% chance of suffering sustained smouldering where they are <93% moisture[41] whilst smouldering fires are capable of horizontal spread in peats below 150% moisture content[42]. A 50% probability of ignition has also been estimated in peat moisture contents between 110% and 125%[43,44]. This implies that the ignition risk of these regionally important carbon stocks[45] in the organic layers at all sites in this study was considerably increased during the July 2022 heatwave with the potential for smouldering wildfires; indeed, peatland carbon emissions were found to be higher during particularly dry years within the UK due to increased frequency of wildfires[46].

While the seasonality of different controls on the moisture content of fuels typically reduce the likelihood of severe wildfires occurring, we have shown that extreme heatwaves, which will become more frequent and intense under climate change[5], can override phenological and landscape controls and the associated ecosystem resistance to large, severe wildfires. Lower moisture contents of all fuel layers during extreme weather aligns the conditions for increased risk of ignition (low dead moisture content), high rates of fire spread (through live *Calluna*) and greater fire severity with the risk of ignition of ground fuels[19,21]. The more extreme fire behaviour and risk of smouldering has the potential to challenge the existing fire adaptations of these plant communities. *Calluna* in particular hosts post-fire vegetative regenerative traits, including vegetative resprouting from stem bases, adventitious roots[47] and fire-stimulated seed germination[48]. In general, *Calluna* can withstand light to moderate severity fires which may top-kill it but do not usually damage stem bases or destroy the soil seedbank[49]. Severe fires, such as those associated with heatwave conditions, may kill seeds[50], destroying the extensive seed banks that are required for sexual reproduction of this species. Therefore, if heatwaves increase fire risk and severity in these ecosystems, effective fuel management strategies will be required to build resilient communities capable of maintaining biodiversity and reducing carbon loss. Fuel management will also become essential to manage the fire that will prevail under extreme conditions because without this, flame lengths and rates of spread will exceed traditional direct attack strategies

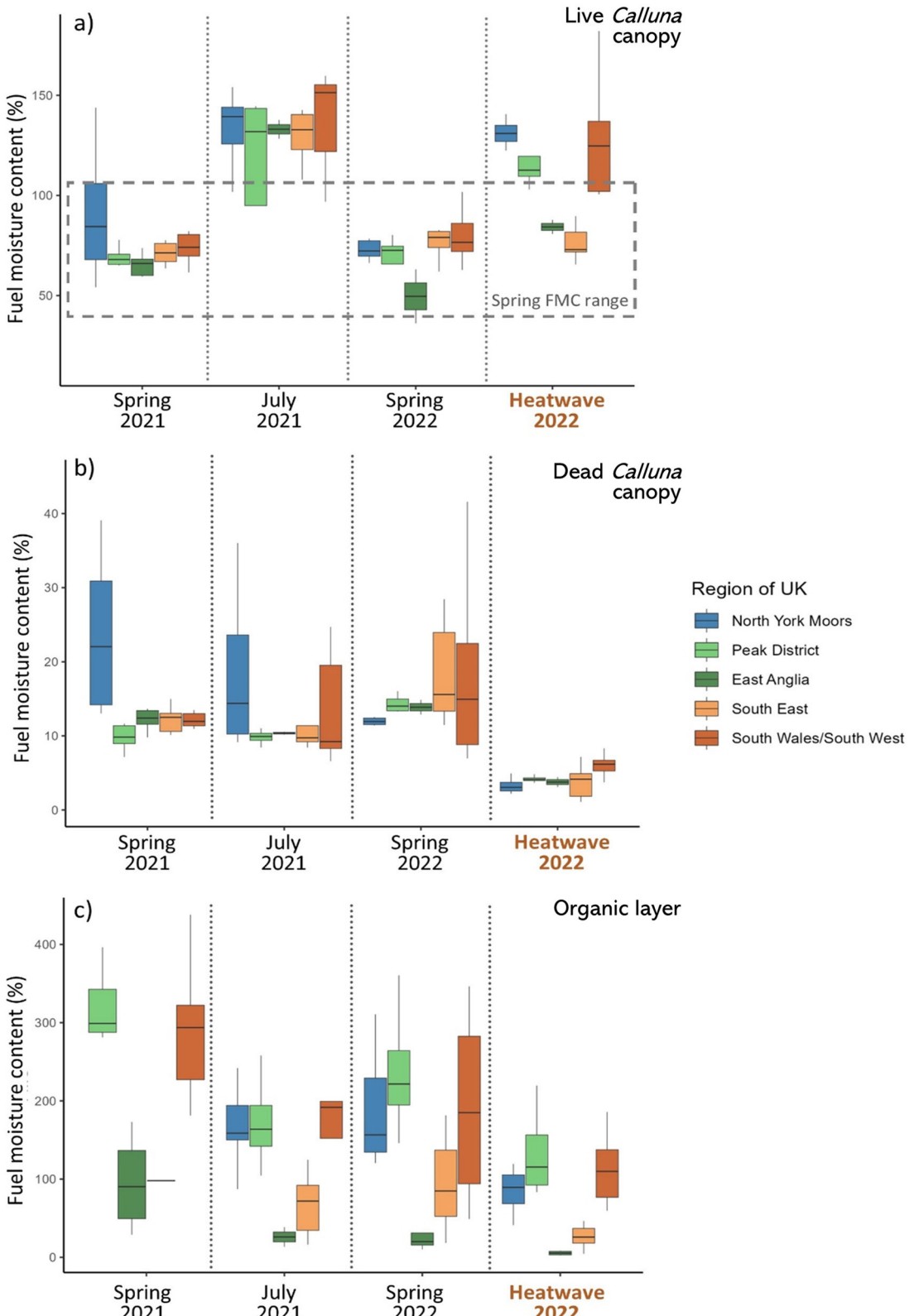

**Fig. 4 | Seasonal patterns of FMC during typical and heatwave conditions.**
Moisture content of **a** live *Calluna* canopy, **b** dead *Calluna* canopy and **c** the organic soil layer during Spring (March–April) 2021, July 2021, Spring (March–April) 2022 and the heatwave of 2022 (17th–19th July) from sample sites in different regions of the UK. In boxplots, centre line shows the median; box limits show the upper and lower quartiles; whiskers show 1.5× the interquartile range. Outliers have not been plotted; outliers are shown in Supplementary Fig. 1.

using hand tools (Supplementary Table 4) and will likely necessitate indirect attack through control lines or advanced aerial direct attack options[51]. Highly flammable summer conditions in heathlands and peatlands will align extreme fire conditions across different temperate ecosystems, increasing the risk of landscape scale wildfires that connect across ecosystem types within fragmented landscapes. We have already witnessed the effect of such an extreme heatwave on wildfire occurrence and severity during July 2022, where there were an unprecedented number of wildfires across the UK resulting in property evacuation and subsequent damage[9], and these events are predicted to become more frequent in the future[5,6]. This emerging threat must be tackled through appropriate landscape management and fire preparedness strategies, such as the allocation of sufficient resources for firefighters and the land management community during such events, that develop alongside these changing fire regimes.

## Methods

### Study region
We established fuel moisture monitoring sites at 43 peatland, heathland and acid grassland locations across Great Britain (Fig. 1b). The monitoring sites were selected to encompass the key fuel types, land cover types[52] and landscape factors that vary within these temperate ecosystems. We endeavoured to represent the combinations of these factors across our sites as much as possible to determine their influence on fuel moisture content. The sites were spread over key precipitation regions in Great Britain[53] and were named as follows: Southwest & South Wales, Southeast, Peak District, East Anglia, the North Yorkshire Moors and the Scottish Highlands.

### Data collection
We collected fuel samples at each site on a fortnightly basis (monthly only over winter) during 2021 and 2022 and more sporadically over 2023 (Supplementary Table 11). In addition to this regular sampling, we completed an intensive sampling campaign during the extreme fire weather period of July 17th–19th 2022, where samples were collected from all regions except the Scottish Highlands (due to wetter conditions) within this three-day period totalling 30, 30 and 47 samples for live *Calluna* canopy, dead *Calluna* canopy and the organic layer, respectively. We collected samples following the sampling protocol of Little et al.[24], modified from Norum and Miller[54]. We collected live and dead *Calluna* canopy material from ca. 10 different plants along a 20 m transect through the site, and the top 5 cm of organic material beneath the surface litter and above the mineral soil at four points along the transect. Only organic material was sampled from coniferous forests. We stored each fuel constituent in an aluminium screw-fit tin sealed with masking tape. We calculated gravimetric fuel moisture content (mass of water per mass of dry sample, %) by weighing the collected samples, drying them for 48 h at 80 °C and then reweighing the dried samples. The FMC is calculated as:

$$\frac{W - D}{D - T} \times 100$$

where $W$ = wet weight (g), $D$ = dry weight (g) and $T$ = the weight of the sample tin (g).

Temperature and relative humidity were largely recorded at each sample site at the time of sampling using a Kestrel weather metre (Kestrel Instruments, Boothwyn, PA). However, some temperature data were missing for samples in early 2021, and relative humidity measurements were missing from the Scottish Highlands dataset. To replace these, we filled in the missing weather data using the Met Office's MIDAS Open hourly weather data[55] and the Environmental Information Data Centre[56] from weather stations close to each sample region (Supplementary Table 5). Using air temperature and relative humidity, we then calculated Vapour Pressure Deficit (VPD). In addition to these, we downloaded daily mean temperature and daily precipitation data at a 0.25° resolution (roughly 27 km at the equator) from the E-OBS ensemble gridded dataset version 26.0[57]. For each sample site, we calculated the long-term mean daily

temperature of the site's associated grid cell for the 20-year period 2004–2023 to represent the long-term climatology of the site and time of year. With the downloaded rainfall data, we calculated the number of days since rain to represent longer-term drying. The sum of total precipitation of the sampling day and the 5 days prior to sampling was calculated to represent shorter-term rainfall information.

Soil types for each sample site were defined using Soilscapes[58], derived from the National Soil Map of England and Wales, and the National Soil Map of Scotland[59]. Soilscapes uses a simplified classification of the national soil map, so Scottish soil types were assigned a Soilscapes classification according to their closest match. We grouped the Soilscapes categories into five groups to aid analysis, using the Soilscape descriptions: loamy and freely draining; sandy and freely draining; loamy and naturally wet; sandy and naturally wet; peaty and naturally wet (Supplementary Table 6). Elevation data were downloaded using the R package 'elevatr'[60], and slope and aspect data were obtained using the package 'raster'[61]. Finally, NDVI values were obtained from the MODIS Vegetation Indices[62] via the Google Earth Engine in 16-day intervals. NDVI was assigned according to which 16-day interval the field sampling date fell into.

### Data analysis
We used a two-stage modelling approach for each of the three fuel types separately to determine the effects of weather, phenology and landscape variables (Table 1) on FMC. We chose a two-stage modelling approach due to the temporal mismatch between (spatiotemporally varying) weather and phenology variables, and the (spatially varying) landscape variables. Firstly, we ran a mixed-effects linear model for FMC with fixed effects comprising weather variables (VPD, number of days since rain and the total rainfall of the sampling day and 5 days prior—hereon called five-day rainfall) and phenological variables (NDVI and long-term mean daily temperature), with site as a random effect. Sample year and time of day were included as fixed effects to account for diurnal and interannual FMC variability. Within these models, we calculated the marginal R-squared values, which tell us the % of FMC variability explained by weather and phenological variables (fixed effects), and the conditional R-squared values, which tell us the % of FMC variability explained by all variables, including the site-level random effects. Estimates calculated in the model depict the strength of the relationship (either positive or negative) of each variable with FMC variability (see Fig. 3). We extracted the random effect for each site and created a second linear model for these effect sizes with landscape variables (soil type, land cover type, elevation, aspect and slope) as fixed effects. We calculated the relative importance of each variable on across-site FMC variability by extracting each variable's $t$-value, which represents the magnitude of the relationship between a predictor variable and the response variable. For categoric variables (soil type, land cover type), the model calculates a $t$-value for each category's subgroup (e.g. for acid grassland, bog, coniferous forest and heathland within land cover). These were summed to give the overall importance for each categoric variable. We extracted the adjusted R-squared value from this landscape model which tells us, of the % of FMC variability explained by site-level effects (i.e. the conditional R-squared—marginal $R$-squared from the first model), what % variability is due to the landscape variables in the second model. This method allowed us to investigate the effects of weather, phenology and landscape while accounting for the fact that weather and phenological variables vary over time, while landscape variables were consistent for each site. See Supplementary Tables 7–9 for model summaries and Supplementary Figs. 2 and 3 for landscape effect plots. The correlation between variables is shown in Supplementary Table 10. We ran all statistical analyses in R version 4.3.0[63].

To determine how the heatwave in July 2022 affected FMC, we compared samples of live and dead *Calluna* canopy and the organic layer taken from spring (March and April) 2021, July 2021, spring 2022 and the heatwave in July 2022 (17th–19th July) using Wilcoxon tests between each pair of time periods for each fuel layer.

**Table 1 | Variables used in the two-stage modelling approach**

| Variable | Units | Type of variable |
|---|---|---|
| Vapour Pressure Deficit (VPD) | hPa | Weather |
| Number of days since rain | Days | Weather |
| Total rainfall of sampling day and 5 days prior | Millimetres | Weather |
| Long-term (20 years) mean daily temperature | ° Celsius | Phenological |
| Normalised Difference Vegetation Index (NDVI) | NDVI | Phenological |
| Time of day | Decimal time (e.g. 12:30 = 12.5) | |
| Sample year | One of 2021, 2022 or 2023 | |
| Land cover type | One of: heathland, acid grassland, bog or coniferous forest | Landscape |
| Soil type | A variety of soil types characterised by Soilscapes (Supplementary Table 6[46]) | Landscape |
| Elevation | Metres | Landscape |
| Slope | Degrees | Landscape |
| Aspect | Degrees (0 = north-facing, 90 = east-facing, 180 = south facing, 270 = west-facing) | Landscape |

## Fire behaviour predictions

In order to estimate the variation in the fire behaviour that might result from heatwave conditions, we used the BehavePlus modelling software[64] (Andrews, 2010). We contrasted mean spring conditions (average FMC of April 2021 and April 2022) and mean July 2021 FMC conditions with those of the heat wave July conditions for the Southeast UK. We chose this region for the modelling because this is the region that suffered loss of homes during the heatwave conditions in July 2022[65]. We ran the surface fire module in BehavePlus and the Probability of Ignition Module. We used a fuel model designed for UK *Calluna* ecosystems[15]. We set the fuel moistures inputs to only require dead and live categories. We kept slope angle (set at 10%) and wind speed (set at 30 km/h) static between the runs and assumed no fuel shading from the sun. The input variables are shown in Supplementary Table 4.

## Reporting summary

Further information on research design is available in the Nature Portfolio Reporting Summary linked to this article.

## Data availability

The data used in this study are available at: https://doi.org/10.6084/m9.figshare.c.7517214. For this study, 'Fuel.layer' column should be subset to include only 'Heather dead canopy', 'Heather live canopy' and 'Duff' (organic layer).

## Code availability

The code used in this study are available at: https://doi.org/10.6084/m9.figshare.c.7517214.

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

## Acknowledgements
Thank you to the land holders and managers for permitting sampling to take place on their land. This project has received funding from NERC Highlight project NE/T003553/1.

## Author contributions

C.M.B., G.D.C., S.H.D., K.L., N.K., A.O. and T.E.L.S. developed the study design; R.A., L.J.G., K.L., N.K. and A.O. collected data; L.J.G., K.I., K.L. and N.K. analysed the data; K.I. and N.K. led the writing of the manuscript; all authors contributed to the manuscript prior to submission.

## Competing interests
The authors declare no competing interests.
