## [Transparent Peer Review file · Communications Earth & Environment]

Unprecedented UK heatwave harmonised drivers of fuel moisture creating extreme temperate wildfire risk

Corresponding Author: Dr Katy Ivison

Version 0:

Decision Letter:

Dear Dr Ivison,

Your manuscript titled "Unprecedented UK heatwave harmonised drivers of fuel moisture creating extreme temperate wildfire risk" has now been seen by 3 reviewers, and we include their comments at the end of this message. They find your work of interest, but some important points are raised. We are interested in the possibility of publishing your study in Communications Earth & Environment, but would like to consider your responses to these concerns and assess a revised manuscript before we make a final decision on publication.

We therefore invite you to revise and resubmit your manuscript, along with a point-by-point response that takes into account the points raised. In particular, please:

- provide a thorough evaluation of model selection and interpretation, alongside a robust climatological analysis of the 2022 heatwave, comparing it to historical events and future projections.

- provide a detailed discussion on fire ecology, such as ecological implications of fuel moisture dynamics during heatwaves.

Please highlight all changes in the manuscript text file.

Please submit your point-by-point responses as a separate file, distinct from your cover letter where you can add responses to the Editors' comments that you do not want to be made available to the reviewers. Word files are preferred. We recommend that any figures, tables or graphs that are included in the response to reviewers are also included in the main article or Supplementary Information.

Please use the following link to submit your revised manuscript, point-by-point response to the referees' comments (which should be in a separate document to any cover letter), a tracked-changes version of the manuscript (as a PDF file) and the completed checklist:

Link Redacted

We hope to receive your revised paper within six weeks; please let us know if you aren't able to submit it within this time so that we can discuss how best to proceed. If we don't hear from you, and the revision process takes significantly longer, we may close your file. In this event, we will still be happy to reconsider your paper at a later date, as long as nothing similar has been accepted for publication at Communications Earth & Environment or published elsewhere in the meantime.

Please do not hesitate to contact us if you have any questions or would like to discuss these revisions further. We look forward to seeing the revised manuscript and thank you for the opportunity to review your work.

Best regards,

Mengze Li, PhD
Editorial Board Member
Communications Earth & Environment

Alireza Bahadori, PhD
Associate Editor
Communications Earth & Environment

EDITORIAL POLICIES AND FORMATTING

Editorial Policy: [Policy requirements](https://www.nature.com/documents/nr-editorial-policy-checklist.pdf) (Download the link to your computer as a PDF.)

- Behavioural and social science
- Ecological, evolutionary & environmental sciences
- Life sciences

<https://www.nature.com/documents/nr-reporting-summary.zip>

Furthermore, please align your manuscript with our format requirements, which are summarized on the following checklist: [Communications Earth & Environment formatting checklist](https://www.nature.com/documents/commsj-phys-style-formatting-checklist-article.pdf)

and also in our style and formatting guide [Communications Earth & Environment formatting guide](https://www.nature.com/documents/commsj-phys-style-formatting-guide-accept.pdf) .

*** DATA: Communications Earth & Environment endorses the principles of the Enabling FAIR data project (<http://www.copdess.org/enabling-fair-data-project/>). We ask authors to make the data that support their conclusions available in permanent, publically accessible data repositories. (Please contact the editor if you are unable to make your data available).

All Communications Earth & Environment manuscripts must include a section titled "Data Availability" at the end of the Methods section or main text (if no Methods). More information on this policy, is available at <http://www.nature.com/authors/policies/data/data-availability-statements-data-citations.pdf>.

If a community resource is unavailable, data can be submitted to generalist repositories such as [figshare](https://figshare.com/) or [Dryad Digital Repository](http://datadryad.org/). Please provide a unique identifier for the data (for example a DOI or a permanent URL) in the data availability statement, if possible. If the repository does not provide identifiers, we encourage authors to supply the search terms that will return the data. For data that have been obtained from publically available sources, please provide a URL and the specific data product name in the data availability statement. Data with a DOI should be further cited in the methods reference section.

REVIEWER COMMENTS:

Reviewer #1 (Remarks to the Author):

General comments:

This paper explores an important topic; targeting the knowledge gap of for fuel-fire relationships in more temperate fire systems. While the paper is intriguing conceptually, I have serious concerns on the analysis and interpretation. It appears that AIC may be being interpreted incorrectly by the authors (choosing models with higher AICs rather than lower AIC values). If so, then the results and interpretations are not the correct ones to make, and cast doubt on the veracity and soundness of the conclusions. I also believe that the claims that live and dead fuel categories differ greatly in the drivers may be overstated; in both, humidity is still far and away the most important driver. Finally, I would like to see better contextualization of the ecological role of fire in these ecosystems, and also what role, if any, fire did actually play during the heat wave. Simulated results did show increased fire danger, but the title makes it sounds like the results show that fires did actually occur—although this is never discussed.

Specific comments:

Abstract: Do you need to include references in the abstract? Don't know style on this specific journal, but it seems uncommon.

50: What is the fire ecology of this species/ecosystem? Are they fire adapted? I'd like to see a little bit more of this set-up, to understand what fire means in these ecosystems.

51: Briefly explain how representative *Calluna vulgaris* is of all heathlands and peatlands in the UK. Are there other studies using this as a model ecosystem; is it the dominant type of heathland/peatland/etc.

66: A conceptual figure may be valuable here to lay out all the different drivers of fuel moisture/fire behavior in these systems.

72 (and throughout): would "seasonal" or "phenological" be a better term than "temporal" to differentiate from weather? When I first read temporal, I already thought about weather, since that it changes so much through time. [Update: I see that you also include time of the day, so seasonal is not appropriate. Instead, I would better layout your categories before launching in so they are not confusing]

99: Did you consider using any lagged variables (e.g., precip of previous 1 day)? I wonder if it's less that these different fuel categories have different drivers and more that the lag of the drivers varies. It makes sense that live fuel is buffered more from the immediate weather since the plant regulates itself actively.

100: As above, I think a preemptive explanation of variable categories would be good. It's confusing why some are temporal vs. weather. For example "Temperature" is a weather variable, but "Mean temperature of sample day" is a temporal variable. It seems like these are very related.

117-123: Where there are actual, recorded fires during this time in this ecosystem?

125: Why just temperature? And not other variables that were more important in the model (e.g., humidity)

128-129: I think this is an overly-strong way to say that it becomes "strongly negative"; am I correctly interpreting your text that p-value of the slope above 30 degrees is 0.07? If using a threshold of $p = 0.05$, this test does not provide evidence for that, so I think the language of "strongly negative" is misleading.

131-132: Lower AIC values are better, so wouldn't the piecewise regression be the better model here? This is also evident in that in the Dead *Calluna* model (supplemental 6), the piecewise regression values are also significant, unlike the Live *Calluna* model.

135-136: Again, the piecewise regression has lower AIC values, indicating it is the better model. I believe this whole section might be choosing the worse-fit models.

Figure 3: I think these model selections might be entirely backwards (piecewise when they should be linear; and vice versa). Make sure to check this, and make sure correct AIC interpretation is being made. For example, why is Dead *calluna* not a piecewise when the Supplemental Table 6 seems to show that this model is significant and a better fit?

170-171: I think it's less that they break down, but rather that that they may change; there are threshold effects going on.

183-185: Again, I think that the 30 degree threshold here is not supported by the model selection.

188: Any variable bounded at 0 like fuel moisture percent would be expected to have less variability (i.e., smaller residuals) when smaller; I don't think the residuals are what supports the statement that "weather replaces landscape factors as the largest driver..."

200-202: Was there any fires? Would love to see more in-depth discussion of this risk of fire. Especially since it was a modeled study, it's even more important to link it to any potential tests of the model (i.e., places that did burn).

214-215: Is the solution fire suppression? I would like to see more discussion of the ecological role, if any, of fire in these ecosystems. Just a little addition to contextualize this, with the knowledge that increased fire suppression in other ecosystems (especially frequent fire ecosystems) has only made the problem worse.

227: Why the change in sampling frequency?

Figure 4: Good coverage of sites spatially! Can highlight this fact in start of paper even more.

Reviewer #2 (Remarks to the Author):

This paper characterizes the drivers of live and dead fuel moisture over several years across a range of climatic zones in the UK, including during a period of extraordinary heat. It shows that different drivers affect live and dead fuel. Typically, live and

dead fuels are temporally misaligned in terms of when they are most flammable in this region, yet during the heatwave they aligned, which carries important implications for the probability and intensity of fire (modelled by the authors). Overall, I found it to be a very nice paper, was well written and enjoyable to read, and I think it should be published after incorporating my fairly minor suggestions.

Abstract: The abstract should explain the focal region of study. I think the abstract is also a bit too general. Specifically, it would be good to know a bit more about what you actually did e.g., took field measurements of live and dead fuel over three years including intensive surveying during a record-breaking heatwave to resolve aim xxxxxx. I also think there could be a clearer statement in the abstract that live and dead fuels are typically misaligned in terms of when they are driest in this region (because live fuel is moister when growing in summer) because that provides the context for why the result is perhaps unexpected.

Variables: The authors use the term “temporal variables” throughout the entire manuscript. This term is not defined, and it doesn’t really make sense to me because the so-called “weather” variables are also temporal in nature, and some of the temporal variables measure weather phenomena. Relative humidity, for example, is classed as a weather variable, whereas mean temperature of the sample day is classed as a temporal variable. This needs clarifying, and my view is that the term temporal variable should be dropped altogether, or at the least for any variable reflecting a weather component. If you intend to describe weather phenomena over different time scales, then I suggest using a descriptor of those time scales. Given almost all variables target a mechanism, it also doesn’t make sense to me to include in the models some variables that do not (such as sample year, which should be captured by the weather variables [year itself is not meaningful]).

Modelling: The linear and piecewise regression models could use some improvement. The key deficiency is that the models assume a normal distribution for data that are by definition non-normally distributed because fuel moisture content cannot be negative. The implications of this are seen in Fig 3b, where the model predicts negative fuel moisture at around 35C, which is impossible. This could be very easily remedied by using an appropriate response distribution – either by log-transforming the response variable or using a gamma-distributed GLM (and then back-transforming for visualising the models). This is necessary for statistical rigour, and it would also likely improve the fit of the curvilinear pattern in Fig 3b and perhaps Fig 3c.

Figures: I think it would be nice if the first figure was a map that contextualises the field study and the heatwave. Specifically, I would find it useful if you showed two or more side-by-side maps, one characterising the context of the UK by showing elevation (which is a variable used in the models and is also correlated with rainfall and temp), and (2) a gridded map characterising the heatwave e.g. mean temp during the heatwave. Also showing typical rainfall would be useful. This could replace Fig 4 which is relatively light on information content and is at the end of the method (too late in the paper to really be useful in understanding the context).

Specific line comments

L78 “norm”: suggest tempering this language for two reasons. (1) From a very quick glimpse, that paper seems to be based on RCP8.5, which is a worst-case scenario that is actually rather unlikely (see Hausfather and Peters 2020 Nature). (2) That paper seems to indicate that while probability of weather similar to an extreme year (1976) is predicted to increase by 5 fold under RCP8.5, it is increasing from a very low baseline probably, so saying the “norm” seems like a stretch.

L88-89: Suggest explaining why live Calluna canopy moisture increases during warmer summer months; e.g., as vegetation starts photosynthesising after winter dormancy, or whatever it is.

L91: Suggest saying “relative humidity” rather than “humidity”, just to be consistent with earlier usage and avoid any potential confusion.

L179: “South East” is usually one word.

L247: days to rainfall might capture drying trend more crudely than a summed measure over a longer time period - e.g. if the rain event was 1 mm, it would have quite a different effect than if the nearest rain even was 20 mm. Did you consider the implications of what constitutes a rainfall event?

L274: when reading the main body, I did wonder why you didn’t use VPD, so I think it would be a worth adding a sentence to the main body explaining that you did look into it.

Table 1. double spacing in makes it a bit difficult to distinguish separate rows from multiline entries. Suggest reformatting more aesthetically.

References

Hausfather, Z. et al. (2020) Emissions—the ‘business as usual’ story is misleading. Nature.

Reviewer #3 (Remarks to the Author):

I have now finished my review of “Unprecedented UK heatwave harmonised drivers of fuel moisture creating extreme temperate wildfire risk” by Ivison et al. The manuscript presents strong evidence that atmospheric conditions driven by an extraordinary heatwave in July 2022 led to synchronous extremely low fuel moisture content (FMC) in both live and dead

Calluna canopies, as well as in the organic soil layer. Normally, the moisture content of these three fuel components are controlled by different drivers, namely the annual cycle (i.e., temporal variables), short-term weather variability, and landscape variables, respectively—and, therefore, the landscape does not usually show high fire risk. Based on field observations and adequate statistical analyses, results supporting the conclusion that said heatwave was responsible for the harmonised very low FMC are presented and discussed. Considering the July 2022 evidence, the conclusion states that summer heatwaves will likely play a more relevant role in the future as drivers of wildfire risk. The manuscript is well structured and written, and the methods section is clear and complete. The statistical analyses applied are correctly justified and sufficient information is provided to reproduce a similar work. In my opinion, this study will be of interest for the scientific community and also for decision makers. However, the manuscript will surely be improved after the following points are considered.

Major comments

- 1) My main concern is the lack of a more detailed description of the weather pattern associated to this extreme heatwave and a historical context of its occurrence. By including this climatological background, a stronger statement on its uniqueness could be made.
- 2) A strength of this study is the field campaign conducted during the July 2022 heatwave, so that FMC observations are available during an exceptional heatwave. I understand that field measurements are scarce, but how does FMC look like during other heatwaves within the whole study period? Do you have any other relatively hot event that supports the main claims of this study?
- 3) The manuscript states that the July 2022 heatwave is “an analogue for future extremes” (line 102). What trends and statistics are expected concerning heatwaves in UK in future climate projections? What is the occurrence probability change of an extreme event similar to the July 2022 heatwave?

Minor comments

line 87 and elsewhere: NDVI is presented as a “temporal variable” that exerts a control on Calluna canopy moisture content—indeed its third most important factor. Although this might be a matter of terminology, clarification is needed concerning the fact that NDVI should be regarded as a phenological manifestation of moisture content rather than a driver of it. A discussion on that should be included.

line 108: For readers who are not familiar with FMC, please introduce briefly how live canopy FMC can reach values above 100%.

Figure 2: The last sentence of the legend reads that outliers have not been plotted. How does the figure including outliers look like?

lines 246-247: How long is the period considered to examine longer-term drying (i.e., number of days since rain)? From Figure S2 it is apparently 5 days, but this should not be referred to as “long-term”.

line 285: Does the piecewise regression try different thresholds sequentially or is it an input parameter of the function? Please clarify.

Supp. Material

Figure S2: please clarify “BRTs have a learning rate of 5 and a learning rate of 0.001.” Does the first parameter refer to tree complexity (Table S9)?

Table S10: Which are the bold values referred to in the table’s description?

Communications Earth & Environment is committed to improving transparency in authorship. As part of our efforts in this

direction, we are now requesting that all authors identified as 'corresponding author' create and link their Open Researcher and Contributor Identifier (ORCID) with their account on the Manuscript Tracking System prior to acceptance. ORCID helps the scientific community achieve unambiguous attribution of all scholarly contributions. You can create and link your ORCID from the home page of the Manuscript Tracking System by clicking on 'Modify my Springer Nature account' and following the instructions in the link below. Please also inform all co-authors that they can add their ORCIDs to their accounts and that they must do so prior to acceptance.

Version 1:

Decision Letter:

Dear Dr Ivison,

Your revised manuscript titled "Unprecedented UK heatwave harmonised drivers of fuel moisture creating extreme temperate wildfire risk" has now been seen by 3 reviewers, whose comments are appended below. You will see that although the reviewers appreciate the effort you put into the revisions, they continue to raise substantial concerns about your analysis, arguments, and the robustness of conclusions. Given that these issues have been raised before and not fully addressed, we therefore regret to inform you that we cannot publish your manuscript in Communications Earth & Environment.

In light of these ongoing concerns, together with our editorial considerations, we are unable to conclude that your study is sufficiently compelling to justify publication in Communications Earth & Environment. Unfortunately, these reservations are sufficiently important to preclude publication of this study in Communications Earth & Environment.

We are sorry that we cannot be more positive on this occasion and thank you for the opportunity to consider your work.

Best regards,

Mengze Li, PhD
Editorial Board Member
Communications Earth & Environment

Alireza Bahadori, PhD
Associate Editor
Communications Earth & Environment
Consulting Editor
Communications Sustainability

Reviewers' comments:

Reviewer #1 (Remarks to the Author):

Responses:

Thank you for this revision and for responding the comments of myself and the other reviewers. I appreciate the main point of the paper; that climate change is making regions more fire prone by removing some of the barriers to fire behaviour (even in locations that have not experience as much fire in the past). I think that this is a useful point to make, however I would like to see more explanation about how this is expected to be different from drivers of fire behaviour in locations where this is more understood (e.g., can you place this in the context of similar research that has occurred in other, more fire-prone locations; and how would we expect fire in the UK to differ, or be similar to, other locations). I also appreciate the attention paid to the revised statistical methods, although I have comments regarding that below. However, I still have concerns about the interpretations drawn from the data and the confusing and, in my opinion, somewhat arbitrary, categorization schema.

Additional comments follow below:

Categorization of variables. I think the categorization of variables is important because it has bearing on the conclusions drawn from Figure 3, one of the central points of the paper.

102-106. I still find it very confusing how temperature is in two different categories: I am not convinced that mean air temperature is a "temporal" variable while air temperature from a single point in time is a "weather" variable. Reviewer 2 also asked about this confusing separation, and I do not feel like a convincing explanation was given.

112: "Mean air temperature and NDVI both represent the vegetation green up through the seasonal cycle". Day of the year would be a much more straightforward representation of vegetation green up than mean air temp. 115: "we acknowledge that mean temperature of sample day and NDVI therefore represent the phenological stage of the plant" – I am not convinced that mean temperature of the sample day represents the phenological stage of the plant; day of the year would seem to be much

more linked.

Language of causation

105: "Temporal variables provide an important control on the live Calluna canopy moisture content": NDVI isn't providing a "control" on moisture content (but other way around). And the other "temporal" variable of mean temperature (2nd most influential variable should still be a weather variable).

Use language of "association"/"correlation" not "control" or "driver". You provide some tempering of this language in 115-116, but need to make sure this language is used throughout. For example in line 86: "We determined the primary controls on the fuel moisture" –NDVI is not a control on fuel moisture, but rather the other way around.

Model concerns

I would like to see how much variables are correlated! I'm sure lots of these variables are highly correlated. This makes it even harder to get at causal relationships as potentially confounding variables and reduce the relative influence of a variable, but that doesn't mean it isn't important mechanistically.

Why not use water balance terms (like VPD as Reviewer 2 recommends); you say it gets rid of the influence of temperature and humidity; but that would seem to be partly the point, in that VPD is a more mechanistic/integrative metric of how plants experience temp/precip. Other papers that you cite (e.g., citation 9) also use VPD

126: Did models with VPD in them perform better? I would like to see justification for why they were not presented, not just that they replaced humidity or temp. In many ways (like the other reviewers pointed out), VPD would be a better, more mechanistic predictor so I would like to see justification for how models using VPD compare to models using temp & humidity.

155-165 analysis: why are other variables not included in these models, like they are in the BRT models? It seems like you already modeled this earlier, so I'm curious why there is a need for two different modeling approaches instead of just using one to accomplish both goals (1. showing relative importance or effect size of predictors; 2. Showing model predictions). Even the comparison between years could be incorporated into the full model with an interaction between year and any relevant model; this would also allow for model selection to test whether an interaction with year improved model fit).

Different drivers of live vs. dead

It makes sense that live fuel moisture changes much more slowly than dead fuels, and has larger patterns of phenological changes. In other words, the scale of FMC change is different between live vs. dead. However, I don't think that Figure 3 provides convincing evidence that the two are being "driven" by wildly different things. If you look at what variables are in the top-5 highest relative influence between live and dead you get essentially the same variables for live and dead (not in order).

Live: Humidity; Temp (mean); Time of day (likely to be heavily correlated with temperature); Rainfall; NDVI

Dead: Humidity; Temp (mean); Temp (point in time); Rainfall; NDVI

Basically, it seems like the main argument (that live and dead fuel are driven by different things) relies on the definitions of categories of weather vs. temporal, but these categories are arbitrary and confusing, and ultimately, both live and dead have virtually the same variables as what is the most important.

Other comments:

72: "with ecohydrological drivers differing from weather dependent controls of dead fuels that underpin fuel moisture models in traditionally fire prone regions". How do they differ here compared to other locations that are more studied? There is also a phenological / seasonal pattern in more-studied locations as well.

155: missing "and" between "temperature" and "relative humidity"

Reviewer #2 (Remarks to the Author):

The authors have done a good job of addressing my previous comments. The addition of Figures 1 and 2 is particularly helpful, and the revised analysis using a GLM is a clear improvement in terms of statistical rigor. I think it is close, however I do feel there are several minor issues remaining.

In particular, I remain very unconvinced by the use of the term "temporal variable" and urge the authors to reconsider it:

- While you have added a definition of it in the Results (e.g., L 102), the term appears in the abstract where it is too vague to be meaningful.
- More importantly, the categorization itself is confusing. All weather variables are inherently temporal—they vary over time and reflect seasonal patterns—yet some are referred to as "weather variables" and others (e.g., mean temperature on the sampling day) are categorized as "temporal variables."
- NDVI does indeed vary temporally, but this is driven by weather and plant phenology, so why not just refer to it as a phenological variable, which can be easily understood.
- The variables grouped under the "temporal" label reflect very different processes occurring at different temporal scales. As such, I don't think this grouping adds clarity or analytical value.
- Moreover, there doesn't appear to be a unifying point or interpretation tied to the grouping of "temporal variables," so removing the label could simplify the paper and make the findings easier to interpret.

Instead, I suggest referring to each variable based on what it actually represents—for example:

- NDVI as a measure of plant phenology,
- Time of day as a sampling characteristic, or a "diel variable"
- Sample year as a statistical control.

Additionally, I'm unsure why two separate variables are included to represent the temperature on the sampling day. I would expect these to be highly correlated.

On a related note, given the overlapping and potentially correlated nature of several predictors, did you test for correlations among all variables? This is important because of its implications on the relative importance results. You say that boosted regression trees can handle highly correlated predictors (L309-310), which I believe is really only true when prediction, as opposed to inference, is the goal. Calculating relative influence with two highly correlated variables in the same model will likely underestimate the importance of those variables (because omission of one of the variables will likely not cause significant loss of explanatory power while the other remains in the model). On this note, the method for calculating the relative influence scores is not clear to me (as these can be based on multiple methods), so please add explanation of that too. (Please correct me if I've missed this info).

Reviewer #3 (Remarks to the Author):

In my opinion, the authors have adequately addressed the points raised in the previous round. Explanations and clarifications have been incorporated to the text and new figures provide a more complete context for the study. Thanks to comments by other reviewers, the methods used for the analysis are now better justified. Therefore, I think this manuscript is now in better shape. I recommend it for publication, provided the following minor comments are considered and incorporated into the text:

-“mean (air) temperature of the sample day”, regarded as a “temporal variable”, is interpreted here as representative of the annual cycle of temperature (as I understand, this is indeed the intention with the “temporal” category of variables). However, it should not be confused/interchanged with “long-term mean temperature of a certain day”, which would be the average of mean temperature for a certain day over many years (e.g. 30, as traditionally used in climatology). The latter variable should be regarded as a temporal variable (though maybe not very informative/useful for the purposes of this study), especially considering that the “mean temperature of the sample day” might exhibit a considerable departure from it. Such anomaly, in contrast with the expected climatological value, could be in fact regarded as the consequence of “effective weather”. Then, shouldn't be “mean (air) temperature of the sample day” considered a weather variable, along with “temperature at the time of sampling”? Please clarify and discuss.

-In line with previous comment, please revise lines 123 and 124: which variables represent seasonal and short-term temperature trends, respectively?

-Gamma GLM are now shown in Figure 5, considering July 2021 and July 2022 separately. I suggest fitting a third, more complete model (and plotting its corresponding curve) for each fuel type using the data from both periods. In my opinion, such models would represent a more general approach to the effective influence of weather (temperature) on the corresponding FMC.

Version 2:

Decision Letter:

Dear Dr Iverson,

Your revised manuscript titled "Unprecedented UK heatwave harmonised drivers of fuel moisture creating extreme temperate wildfire risk" has now been seen by our reviewers, whose comments appear below. In light of their advice we are delighted to say that we are happy, in principle, to publish a suitably revised version in Communications Earth & Environment.

We therefore invite you to revise your paper one last time to address the remaining concerns of our reviewers. At the same time we ask that you edit your manuscript to comply with our format requirements and to maximise the accessibility and therefore the impact of your work.

EDITORIAL REQUESTS:

*****Please take care to match our formatting and policy requirements. We will check revised manuscript and return

manuscripts that do not comply. Such requests will lead to delays. *****

SUBMISSION INFORMATION:

OPEN ACCESS:

Communications Earth & Environment is a fully open access journal. Articles are made freely accessible on publication. For further information about article processing charges, open access funding, and advice and support from Nature Portfolio, please visit <https://www.nature.com/commsenv/open-access>

Link Redacted

Best regards,

Mengze Li, PhD
Editorial Board Member
Communications Earth & Environment

Alireza Bahadori, PhD
Associate Editor
Communications Earth & Environment
Consulting Editor
Communications Sustainability

REVIEWERS' COMMENTS:

Reviewer #2 (Remarks to the Author):

This is the third time I have reviewed this manuscript. I thank the authors for making these considered changes. I think these revisions have done a good job of addressing my concerns, some of which were also shared by the other reviewers. I find the revised grouping of variables both more logical and easy to digest. On the whole, I'm pleased to say that I believe the work warrants and is ready for publication.

Reviewer #3 (Remarks to the Author):

In my opinion and regarding the current submission, the authors have covered the main issues raised by the reviewers. In particular, the new classification of variables is more adequate. Nonetheless, the methodological changes incorporated in each version of the manuscript make this review process not easy to follow. All in all, the fact that different (and improved) methods lead to similar results is a strength of this study, and the more direct methodological approach followed this time is welcome. However, it should be described clearly and in detail, in particular regarding the points that I describe below. In other words, a careful presentation of the current shape of the study should be prepared.

In this context, my main concerns are the following:

-Modelling approach: please briefly introduce the "mixed effects model" following the author guidelines for this journal.

Clarify whether there is a linear model implied and how the “estimates” corresponding to each variable (Fig. 3) should be interpreted.

-Definition and clarification of terms: in line with previous comment, particular attention should be paid to the results shown in Figure 3. The concept “effect size” for panels a,b,d just appears in this caption, and is referred to as “estimate” in the corresponding graphs. All terms used in panels and captions should be properly defined in the main text. Moreover and specifically, where is the exact definition of the “variable importance” quantified by t-values in Fig. 3c,e? Without any further methodological details, no clue is provided to the reader for the interpretation of these results.

Panels in Fig. 3c,e are even more delicate, since no indication is given to interpret the meaning of the labels. The caption even indicates “LC”. Again, all terms should be clearly defined in the main text and, in this case, also in Table 1.

-Other comments:

Figure S2: Please describe in the caption all features for each panel. I am particularly curious on the fitted models for elevation, slope, and aspect. Same for similar figures.

** Visit Nature Portfolio's author and referees' website at www.nature.com/authors for information about policies, services and author benefits**

*** We are grateful to the three reviewers for their constructive feedback. Our responses to reviewers' comments are detailed below. Line numbers in the reviewers' comments refer to those in the original manuscript and those in our responses refer to the ones in the revised manuscript with changes. Our responses to each point below marked with '***'. In the manuscript text, our edits are highlighted in yellow.

Reviewer #1:

General comments:

This paper explores an important topic; targeting the knowledge gap of for fuel-fire relationships in more temperate fire systems. While the paper is intriguing conceptually, I have serious concerns on the analysis and interpretation. It appears that AIC may be being interpreted incorrectly by the authors (choosing models with higher AICs rather than lower AIC values). If so, then the results and interpretations are not the correct ones to make, and cast doubt on the veracity and soundness of the conclusions. I also believe that the claims that live and dead fuel categories differ greatly in the drivers may be overstated; in both, humidity is still far and away the most important driver. Finally, I would like to see better contextualization of the ecological role of fire in these ecosystems, and also what role, if any, fire did actually play during the heat wave. Simulated results did show increased fire danger, but the title makes it sounds like the results show that fires did actually occur—although this is never discussed.

*** We thank Reviewer 1 for their comments. We have addressed each comment below in turn.

*** **Piecewise regression models:** We thank the reviewer for highlighting our mistake with the AIC. Based on this, and other subsequent comments about the piecewise regression, we have removed this analysis completely and replaced it with a series of GLMs using Gamma-distribution, which was recommended by Reviewer 2. We have run these for both temperature and relative humidity, based on Reviewer 1's later comment in this document. For the three fuel types we ran GLMs for both July 2021 & July 2022 data combined, as well as separately for July 2021 and July 2022. For live *Calluna* and the organic layer, the relationship between FMC and temperature/relative humidity is significant for the combined timed periods and July 2022 only. It is not significant for July 2021. For dead *Calluna*, the relationship between FMC and temperature/relative humidity is significant for all models. This supports our result that dead *Calluna* FMC is controlled mainly by the weather, as the relationship is significant for all combinations of data. In comparison, for live *Calluna* and the organic layer the influence of weather only becomes significant (a significant relationship between FMC and temperature/relative humidity is only visible) under extreme heatwave conditions.

*** To explained the revised statistical analysis, we have added the following to the methods on line 331 of the revised manuscript: 'We also carried out General Linear Models (GLMs) with Gamma-distribution to test the relationship between FMC and both temperature and relative humidity for (i) July 2021, (ii) all of July 2022 and (iii) both July 2021 and July 2022.'

*** Further, we have added the following to the results on line 155: 'We tested the relationship between FMC and both temperature relative humidity for July 2021 and July 2022 (Fig. 5). For both temperature and relative humidity, when data from 2021 and 2022 was considered together, this relationship was significant for all three fuel types ($p < 0.01$ for all fuels for both temperature and relative humidity). The relationship was also significant for all three fuel types when only July 2022 was considered (temperature model: 0.01, < 0.01 & 0.01 for live *Calluna*,

dead *Calluna* and the organic layer respectively; relative humidity model: 0.01, <0.01 & <0.01 for live *Calluna*, dead *Calluna* and the organic layer respectively). However, when only July 2021 was considered, this relationship was only significant for dead *Calluna* (temperature model: $p = 0.69$, <0.01 & 0.18 for live *Calluna*, dead *Calluna* and the organic layer respectively; relative humidity model: $p = 0.99$, <0.01 & 0.94 for live *Calluna*, dead *Calluna* and the organic layer respectively). For full summaries see Supplementary Tables 6 & 7.'

*** We have revised Figure 5 in line with this updated analysis:

Figure 5: The relationship between fuel moisture content and temperature of live *Calluna* canopy, dead *Calluna* canopy and the organic ground fuel layer during July 2021 and July 2022. Lines show Gamma-distributed Generalised Linear Model relationships between FMC and temperature during each time-period. Significant relationships are solid lines, non-significant relationships are dashed lines. For full summaries of regression models see Supplementary Table 6.

*** Within the discussion we subsequently state:

*** 'During the 2022 heatwave the moisture content of the dead *Calluna* canopy was extremely low. This low moisture content provides a highly ignitable fuel load, evidenced by the 87% probability of ignition (Supplementary 5). Whilst extreme, this was not surprising as our analysis highlights the importance of short-term weather variables, as opposed to temporal or landscape variables, on dead *Calluna* moisture. The significant relationship between FMC and both temperature and relative humidity (Fig. 5, Supplementary Tables 6 & 7) for heatwave and non-heatwave conditions further supports this.' [Line 197; edited section is final sentence]

*** ‘We found that there was a significant relationship between live *Calluna* FMC and both temperature and relative humidity during July 2022 but not July 2021, suggesting that a response of the live fuel moisture content was only evident under the extreme weather seen during the heatwave.’ [Line 207]

*** ‘Similarly to live *Calluna*, we found that the relationship between FMC and both temperature and humidity for the organic layer was again only significant under extreme heatwave conditions (Fig. 5).’ [Line 216]

*** **Differing drivers between live and dead:** The reviewer is correct that relative humidity has the highest influence on live FMC. However, we wanted to highlight the phenological aspect as this creates a very different annual cycle in FMC compared to dead FMC; so, although the main driver of live FMC is relative humidity, there is still a resistance to homogeneously low FMC.

*** We have edited the following on line 175: ‘Within peatlands and heathlands, during periods of high fire weather, notably during the summer, dead fuels are dry and fire ignitions are more probable¹⁷. However, during this period, the moisture content of the live canopy is high²⁵ (Fig. 4a), despite live and dead FMC sharing the same main driver of relative humidity’.

*** We have also added the following on line 211 to address the fact that, although relative humidity has a large influence on live FMC, this relationship was not significant in the GLMs described above. ‘We acknowledge that despite relative humidity having the greatest influence on live FMC, Supplementary Fig. 4 shows that this relationship is strongest above 80% relative humidity; the lack of significance between relative humidity and FMC in July 2021 may therefore be explained as humidity would be unlikely to exceed 80% during the summer.’

*** **Ecological role of fire in these ecosystems:** This is addressed fully in the second specific comment (concerning line 50) where we give some more information about the species, ecosystem and adaptation to fire. In terms of what role fire had in these ecosystems during the heatwave, we have added the following to the discussion on line 241: ‘In general, *Calluna* can withstand light to moderate severity fires which may top-kill heather but do not usually damage stem bases or destroy the soil seedbank⁴³. Severe fires, such as those associated with heat wave conditions, may kill seeds⁴⁴, destroying the extensive seed banks that are required for sexual reproduction of this species. Therefore, if heat waves increase fire risk and severity in these ecosystems, effective fuel management strategies will be required to build resilient communities capable of maintaining biodiversity and reducing carbon loss.’

*** **Fires during the heatwave:** We have added the following to the introduction on line 53: ‘The UK July 2022 heatwave coincided with an unprecedented number of wildfires, overwhelming fire services and causing damage to many properties⁹’. We have added the following to the discussion on line 253: ‘We have already witnessed the effect of such an extreme heatwave on wildfire occurrence and severity during July 2022, where there were an unprecedented number of wildfires across the UK resulting in property evacuation and subsequent damage⁹’

Specific comments:

Abstract: Do you need to include references in the abstract? Don’t know style on this specific journal, but it seems uncommon.

*** We have removed references from the abstract.

50: What is the fire ecology of this species/ecosystem? Are they fire adapted? I'd like to see a little bit more of this set-up, to understand what fire means in these ecosystems.

** We have added the following to the introduction on line 66 to give some context about the species of interest (*Calluna*) and the heathland ecosystems: 'In the UK, such heathlands and peatlands are often characterised by the dwarf shrub *Calluna vulgaris* (hereafter *Calluna*), a fire-adapted species which regenerates after burning¹⁴, though regeneration is poorer within older *Calluna* plants¹⁵. Peat is a particularly good substrate for *Calluna* seedling growth following fire¹⁶. Many contributions to UK wildfire science have focused on *Calluna*-dominated heathlands and peatlands (e.g. Davies et al.¹⁵, Davies and Legg¹⁷, Davies et al.¹⁸).'

*** We have also discussed the possible repercussions of severe fires within these ecosystems – please see our response to your general comments above.

51: Briefly explain how representative *Calluna vulgaris* is of all heathlands and peatlands in the UK. Are there other studies using this as a model ecosystem; is it the dominant type of heathland/peatland/etc.

*** Please see our response to the comment directly above this and the response to the general comments which give some more information about the ecosystem and mentions that this type of ecosystem has been researched before in the wildfire context.

66: A conceptual figure may be valuable here to lay out all the different drivers of fuel moisture/fire behavior in these systems.

*** We have added a figure (Fig. 2, below) which outlines the patterns/drivers of live, dead and organic FMC within these ecosystems.

Figure 2: Drivers of fuel moisture content (FMC) of live *Calluna*, dead *Calluna* and the organic soil layer. Live FMC is lowest in spring and increases in summer as green leaves emerge. Dead FMC is driven by short-term weather, increasing in wet conditions and decreasing in warm, dry conditions. Organic FMC is influenced by water retention and soil drainage so is at least partly driven by weather conditions, but this effect may be delayed.

72 (and throughout): would “seasonal” or “phenological” be a better term than “temporal” to differentiate from weather? When I first read temporal, I already thought about weather, since that it changes so much through time. [Update: I see that you also include time of the day, so seasonal is not appropriate. Instead, I would better layout your categories before launching in so they are not confusing]

*** We have added the following to the Results section on line 102 to make it clear how we have defined each variable type: ‘Here, we used ‘temporal’ to refer to variables that represent any aspect of time, including time of day, season or phenology (including mean temperature of the sample day, as this is indicative of the time of year) and year. ‘Weather’ refers to meteorological conditions such as rainfall, humidity and temperature at the time of sampling. Finally, ‘landscape’ refers to physical characteristics of a sample site, such as elevation, aspect, soil type.’

99: Did you consider using any lagged variables (e.g., precip of previous 1 day)? I wonder if it’s less that these different fuel categories have different drivers and more that the lag of the drivers varies. It makes sense that live fuel is buffered more from the immediate weather since the plant regulates itself actively.

*** We understand that the lag length may affect the importance of a variable on the different fuel types. However, we wanted to include the same variables for each fuel type to allow for comparison of drivers between fuels, and were conscious of including too many variables (e.g. if we had included 1 day lag, 3 day lag, 5 day lag etc as three separate variables). Figure S2

presents our analysis carried out to choose how many lagged days to include in the model, and shows that lagged rainfall information does influence FMC more than rainfall on just the day of sampling for live Calluna. The models also show that lagged rainfall has a slightly higher influence on dead than live Calluna. However, we did not test different lags on each fuel type to look at whether the drivers differ – we wanted to create consistent models across fuel types so that we could easily compare the relative influence of each variable on FMC. We were also conscious of including too many similar variables in the model (e.g. if we had included 1 day lag, 3 day lag, 5 day lag etc as three separate variables).

100: As above, I think a preemptive explanation of variable categories would be good. It's confusing why some are temporal vs. weather. For example "Temperature" is a weather variable, but "Mean temperature of sample day" is a temporal variable. It seems like these are very related.

*** Please see our response above (two comments above this concerning line 72).

117-123: Where there are actual, recorded fires during this time in this ecosystem?

*** Please see our response to the general comments above for our response to this.

125: Why just temperature? And not other variables that were more important in the model (e.g., humidity)

*** We have run GLMs for humidity and added this into the results section; details are given in response to the first general comment above. We have added the results table to supplementary beside the results table for the temperature models but have chosen not to present a figure for the humidity model. This is because we felt that temperature was the more accessible variable of the two, and were concerned about having too many figures in the manuscript. However, we can add a figure for the humidity models if the reviewer thinks this would be valuable.

128-129: I think this is an overly-strong way to say that it becomes "strongly negative"; am I correctly interpreting your text that p-value of the slope above 30 degrees is 0.07? If using a threshold of $p = 0.05$, this test does not provide evidence for that, so I think the language of "strongly negative" is misleading.

*** Please see response above where we have replaced piecewise regression models with Gamma-distributed GLMs.

131-132: Lower AIC values are better, so wouldn't the piecewise regression be the better model here? This is also evident in that in the Dead Calluna model (supplemental 6), the piecewise regression values are also significant, unlike the Live Calluna model.

*** Please see response above where we have replaced piecewise regression models with Gamma-distributed GLMs.

135-136: Again, the piecewise regression has lower AIC values, indicating it is the better model. I believe this whole section might be choosing the worse-fit models.

*** Please see response above where we have replaced piecewise regression models with Gamma-distributed GLMs.

Figure 3: I think these model selections might be entirely backwards (piecewise when they should be linear; and vice versa). Make sure to check this, and make sure correct AIC interpretation is being made. For example, why is Dead calluna not a piecewise when the Supplemental Table 6 seems to show that this model is significant and a better fit?

*** Apologies for this error. We have replaced this Figure (now Figure 5) which now shows the results from our Gamma-distributed GLMs. The Gamma distribution is the most appropriate distribution for our data, which can only take positive values, as pointed out by reviewer 2

170-171: I think it's less that they break down, but rather that that they may change; there are threshold effects going on.

*** Please see response above where we have replaced piecewise regression models with Gamma-distributed GLMs.

183-185: Again, I think that the 30 degree threshold here is not supported by the model selection.

*** Please see response above where we have replaced piecewise regression models with Gamma-distributed GLMs.

188: Any variable bounded at 0 like fuel moisture percent would be expected to have less variability (i.e., smaller residuals) when smaller; I don't think the residuals are what supports the statement that "weather replaces landscape factors as the largest driver..."

*** We have removed this and now refer just to the fact that the relationship between FMC and temperature for the organic layer was only significant during July 2022 (line 216): 'Similarly to live Calluna, we found that the relationship between FMC and both temperature and relative humidity for the organic layer was again only significant under extreme heatwave conditions (Fig. 5).'

200-202: Was there any fires? Would love to see more in-depth discussion of this risk of fire. Especially since it was a modeled study, it's even more important to link it to any potential tests of the model (i.e., places that did burn).

*** Please see our response to the general comments above for our response to this.

214-215: Is the solution fire suppression? I would like to see more discussion of the ecological role, if any, of fire in these ecosystems. Just a little addition to contextualize this, with the knowledge that increased fire suppression in other ecosystems (especially frequent fire ecosystems) has only made the problem worse.

*** We have added some context about fire within the heathland/peatland ecosystem in response to your first general comment above. In terms of the solution to the increased risk of fire frequency/intensity in the future, we acknowledge that there is a lot of discussion around how to manage fires (e.g. suppression, managed burns during low-risk seasons). A discussion of the potential solutions to the risk highlighted are beyond the scope of the paper presented.

However, we have edited the concluding sentences on line 256: 'This emerging threat must be tackled through appropriate landscape management and fire preparedness strategies, such as the allocation of sufficient resources for firefighters and the land management community during such events, that develop alongside these changing fire regimes.'

*** We have also added this sentence on line 244 to touch on this topic: 'Therefore, if heat waves increase fire risk, behaviour and severity in these ecosystems, effective fuel management strategies will be required to build resilient communities capable of maintaining biodiversity and reducing carbon loss.'

227: Why the change in sampling frequency?

*** Fewer samples were taken in 2023 due to staffing changes so there was less availability for samplers to go out into the field as frequently.

Figure 4: Good coverage of sites spatially! Can highlight this fact in start of paper even more.

*** Thank you. We have edited the following to highlight the spatial coverage of our sites on line 88: '5,845 fuel moisture samples were collected from 43 sampling sites (Fig. 1b) across five different climate regions of the UK, from the Scottish Highlands (58.4 °N) to the South West of England (51.1 °N), over a period of three years (2021-2023).'

*** We have also edited the sample sites map figure and moved it earlier in the text as recommended by reviewer 2.

Reviewer #2:

This paper characterizes the drivers of live and dead fuel moisture over several years across a range of climatic zones in the UK, including during a period of extraordinary heat. It shows that different drivers affect live and dead fuel. Typically, live and dead fuels are temporally misaligned in terms of when they are most flammable in this region, yet during the heatwave they aligned, which carries important implications for the probability and intensity of fire (modelled by the authors). Overall, I found it to be a very nice paper, was well written and enjoyable to read, and I think it should be published after incorporating my fairly minor suggestions.

Abstract: The abstract should explain the focal region of study. I think the abstract is also a bit too general. Specifically, it would be good to know a bit more about what you actually did e.g., took field measurements of live and dead fuel over three years including intensive surveying during a record-breaking heatwave to resolve aim xxxxxx. I also think there could be a clearer statement in the abstract that live and dead fuels are typically misaligned in terms of when they are driest in this region (because live fuel is moister when growing in summer) because that provides the context for why the result is perhaps unexpected.

*** We have added the following to the abstract on line 24 to give more information about the study: 'We took field measurements of live and dead *Calluna vulgaris* and organic soil moisture content across the UK over three years, including an intensive sampling campaign during the July 2022 heatwave to investigate the effects of extreme conditions on fuel moisture content.'

*** We have added the following to the abstract on line 30: 'live fuels are drier in spring before the summer green-up, whereas dead fuels are drier in summer due to warmer and drier weather conditions.'

Variables: The authors use the term "temporal variables" throughout the entire manuscript. This term is not defined, and it doesn't really make sense to me because the so-called "weather" variables are also temporal in nature, and some of the temporal variables measure weather phenomena. Relative humidity, for example, is classed as a weather variable, whereas mean temperature of the sample day is classed as a temporal variable. This needs clarifying, and my view is that the term temporal variable should be dropped altogether, or at the least for any variable reflecting a weather component. If you intend to describe weather phenomena over different time scales, then I suggest using a descriptor of those time scales. Given almost all variables target a mechanism, it also doesn't make sense to me to include in the models some variables that do not (such as sample year, which should be captured by the weather variables [year itself is not meaningful]).

*** Based on this comment and that of another reviewer, we have defined each of the three variable groups in the results section (line 102): 'Here, we used 'temporal' to refer to variables that represent any aspect of time, including time of day, season or phenology (including mean temperature of the sample day, as this is indicative of the time of year) and year. 'Weather' refers to meteorological conditions such as rainfall, humidity and temperature at the time of sampling. Finally, 'landscape' refers to physical characteristics of a sample site, such as elevation, aspect, soil type.'

*** In terms of year, we agree that it is potentially less meaningful than other temporal variables; however, this was important to include as it accounts for the interannual variation in FMC which may be explained by more than simply differences in weather across years.

Modelling: The linear and piecewise regression models could use some improvement. The key deficiency is that the models assume a normal distribution for data that are by definition non-normally distributed because fuel moisture content cannot be negative. The implications of this are seen in Fig 3b, where the model predicts negative fuel moisture at around 35C, which is impossible. This could be very easily remedied by using an appropriate response distribution – either by log-transforming the response variable or using a gamma-distributed GLM (and then back-transforming for visualising the models). This is necessary for statistical rigour, and it would also likely improve the fit of the curvilinear pattern in Fig 3b and perhaps Fig 3c.

*** Thank you for this suggestion. Based on this and concerns by Reviewer 1, we have replaced the piecewise regression models with Gamma-distributed GLMs. Please see our response underneath the 'General comments' by Reviewer 1 for details.

Figures: I think it would be nice if the first figure was a map that contextualises the field study and the heatwave. Specifically, I would find it useful if you showed two or more side-by-side maps, one characterising the context of the UK by showing elevation (which is a variable used in the models and is also correlated with rainfall and temp), and (2) a gridded map characterising the heatwave e.g. mean temp during the heatwave. Also showing typical rainfall would be useful. This could replace Fig 4 which is relatively light on information content and is at the end of the method (too late in the paper to really be useful in understanding the context).

*** Thank you for this suggestion. We have edited Figure 4 (sample sites) and moved it earlier in the text. This is now Figure 1 (see below) and shows a panel with a mean temperature map during the 19th July 2022 (main heatwave day) and a panel showing an elevation map with sample sites marked.

Figure 1: (a) Mean temperature of main heatwave day (19th July 2022) across the UK, (b) distribution of sample sites and their associated climate regions. Map sources: Esri, DeLorme, HERE, TomTom, Intermap, increment P Corp., GEBCO, USGS, FAO, NPS, NRCAN, GeoBase, IGN, Kadaster NL, Ordnance Survey, Esri Japan, METI, Esri China (Hong Kong), swisstopo, MapmyIndia, and the GIS User Community. Temperature data is from the E-OBS ensemble gridded dataset version 26.0⁵⁰.

Specific line comments

L78 “norm”: suggest tempering this language for two reasons. (1) From a very quick glimpse, that paper seems to be based on RCP8.5, which is a worst-case scenario that is actually rather unlikely (see Hausfather and Peters 2020 Nature). (2) That paper seems to indicate that while probability of weather similar to an extreme year (1976) is predicted to increase by 5 fold under RCP8.5, it is increasing from a very low baseline probably, so saying the “norm” seems like a stretch.

*** Thank you for this information – we have changed the wording from ‘norm’ to ‘more frequent’ (line 96)

L88-89: Suggest explaining why live *Calluna* canopy moisture increases during warmer summer months; e.g., as vegetation starts photosynthesising after winter dormancy, or whatever it is.

*** We have added 'as new leaves grow which have a higher moisture content' to line 114 clarify why moisture content increases with vegetation green up in the summer.

L91: Suggest saying "relative humidity" rather than "humidity", just to be consistent with earlier usage and avoid any potential confusion.

*** We have changed this to relative humidity.

L179: "South East" is usually one word.

*** We have changed this to Southeast throughout and changed South West to Southwest.

L247: days to rainfall might capture drying trend more crudely than a summed measure over a longer time period - e.g. if the rain event was 1 mm, it would have quite a different effect than if the nearest rain event was 20 mm. Did you consider the implications of what constitutes a rainfall event?

*** We did discuss this point when designing our models. We decided that by using the 5 days prior to rainfall and the number of days since rain, we would capture much of the important magnitudinal and temporal characteristics of rainfall events. We accepted in our original discussions that this does not fully represent all of the dynamics of rainfall events, i.e. the size of rainfall events 6 days and more before sampling; however, we felt that this would be less likely to impact FMC than more recent rainfall events.

L274: when reading the main body, I did wonder why you didn't use VPD, so I think it would be a worth adding a sentence to the main body explaining that you did look into it.

*** We have added the following sentence to the results on line 126: 'Models were also created using vapour pressure deficit (VPD) instead of relative humidity and temperature at the time of sampling; in these models, VPD replaced either relative humidity or temperature in terms of relative influence so are not presented in the main text (see *Methods; Data Analysis*).'

Table 1. double spacing in makes it a bit difficult to distinguish separate rows from multiline entries. Suggest reformatting more aesthetically.

*** We have reformatted the table to reduce line spacing and have added a border to distinguish rows more easily.

References

Hausfather, Z. et al. (2020) Emissions—the 'business as usual' story is misleading. *Nature*.

Reviewer #3:

I have now finished my review of "Unprecedented UK heatwave harmonised drivers of fuel

moisture creating extreme temperate wildfire risk” by Ivison et al. The manuscript presents strong evidence that atmospheric conditions driven by an extraordinary heatwave in July 2022 led to synchronous extremely low fuel moisture content (FMC) in both live and dead Calluna canopies, as well as in the organic soil layer. Normally, the moisture content of these three fuel components are controlled by different drivers, namely the annual cycle (i.e., temporal variables), short-term weather variability, and landscape variables, respectively—and, therefore, the landscape does not usually show high fire risk. Based on field observations and adequate statistical analyses, results supporting the conclusion that said heatwave was responsible for the harmonised very low FMC are presented and discussed. Considering the July 2022 evidence, the conclusion states that summer heatwaves will likely play a more relevant role in the future as drivers of wildfire risk.

The manuscript is well structured and written, and the methods section is clear and complete. The statistical analyses applied are correctly justified and sufficient information is provided to reproduce a similar work.

In my opinion, this study will be of interest for the scientific community and also for decision makers. However, the manuscript will surely be improved after the following points are considered.

*** We thank reviewer 3 for their positive feedback.

Major comments

1) My main concern is the lack of a more detailed description of the weather pattern associated to this extreme heatwave and a historical context of its occurrence. By including this climatological background, a stronger statement on its uniqueness could be made.

*** We have added the following to the introduction on line 42: ‘In July 2022, the UK experienced an unprecedented extreme heatwave as a result of a ‘heat dome’, where high pressure pushes warm air downwards and traps it at the surface. Temperatures exceeded 40 °C for the first time ever and the UK Met Office issued a red warning for extreme heat³. In fact, this event was ranked the highest intensity July heatwave since records began in 1878 based on mean and maximum temperatures and broke temperature records across the country⁴.’

2) A strength of this study is the field campaign conducted during the July 2022 heatwave, so that FMC observations are available during an exceptional heatwave. I understand that field measurements are scarce, but how does FMC look like during other heatwaves within the whole study period? Do you have any other relatively hot event that supports the main claims of this study?

*** There was a heatwave in July 2021 in the UK (Met Office issues first UK extreme heat warning - BBC News), but this was confined to the west/southwest and temperatures did not exceed 32 °C. Figure 3 does not show any strong effect of the July 2021 heatwave on FMC, even in the southwest where the heatwave was most prevalent – in fact, FMC in the southwest is generally higher than other regions and this was also the case during July 2021. Furthermore, the temperatures seen in 2021 are observed frequently in today’s climate (e.g. the highest UK temperatures were 34.8 and 33.5 deg Celsius in 2024 and 2023 respectively according to the Met Office). For these reasons, we did not consider this heatwave to be an extreme event. The 2022 heatwave did present a unique opportunity for us to observe the effects of extreme heat on FMC, but also meant that we were unable to compare the patterns observed with other

equally extreme heatwave events.

3) The manuscript states that the July 2022 heatwave is “an analogue for future extremes” (line 102). What trends and statistics are expected concerning heatwaves in UK in future climate projections? What is the occurrence probability change of an extreme event similar to the July 2022 heatwave?

*** We have added to this sentence on line 47: ‘Human-induced climate change is anticipated to intensify heatwaves⁵, and Europe has experienced four record-breaking heatwaves since 2003¹. It is therefore likely that such events are likely to become more frequent and severe in the future; in the UK, the number of heatwave days are predicted to increase by up to 2 days every decade⁶.

*** We have also added the following to the discussion on line 207 (following our discussion of the lowest FMC being observed in the southeast): ‘The Southeast of England is where the greatest increase in heatwave events is predicted to occur⁶, making this result even more significant in terms of future risk.’

Minor comments

line 87 and elsewhere: NDVI is presented as a “temporal variable” that exerts a control on Calluna canopy moisture content—indeed its third most important factor. Although this might be a matter of terminology, clarification is needed concerning the fact that NDVI should be regarded as a phenological manifestation of moisture content rather than a driver of it. A discussion on that should be included.

*** This is an interesting comment and we have added the following to the results on line 115 to address this: ‘we acknowledge that mean temperature of sample day and NDVI therefore represent the phenological stage of the plant and are not acting as drivers of FMC themselves’

line 108: For readers who are not familiar with FMC, please introduce briefly how live canopy FMC can reach values above 100%.

*** We have added the gravimetric FMC formula to the methods and added the following to line 138 where FMC is shown as being above 100%: ‘FMC is calculated as the mass of water per mass of dry sample and can therefore exceed 100%’

Figure 2: The last sentence of the legend reads that outliers have not been plotted. How does the figure including outliers look like?

*** We plotted without outliers because for dead Calluna, the outliers squashed the bars and made the figure difficult to view. We have added the plots with outliers to supplementary (Figure S3) and referenced it in the legend of Figure 2: ‘outliers are shown in Supplementary Fig. 3.’

lines 246-247: How long is the period considered to examine longer-term drying (i.e., number of days since rain)? From Figure S2 it is apparently 5 days, but this should not be referred to as “long-term”.

*** Figure S2 concerns the other rainfall variable, total (cumulative) rainfall of the 5 days prior to sampling. The analysis in this figure shows that the relative influence of rainfall increases as more days prior to the sampling day are included. However, the longer-term drying refers to the number of days since rain variable, which will show whether there has been any drought before sampling (e.g. if number of days since rain is high). We have made a small edit to the text to clarify this on line 290: 'The sum of total precipitation of the sampling day and the five days prior to sampling was calculated to represent shorter-term rainfall information (Supplementary Fig. 2).'

line 285: Does the piecewise regression try different thresholds sequentially or is it an input parameter of the function? Please clarify.

*** Based on concerns by reviewers 1 & 2, we have replaced the piecewise regression models with Gamma-distributed GLMs. Please see our response underneath the 'General comments' by reviewer 1 for details.

Supp. Material

Figure S2: please clarify "BRTs have a learning rate of 5 and a learning rate of 0.001." Does the first parameter refer to tree complexity (Table S9)?

*** Thank you for pointing this out – we did mean tree complexity and have changed this.

Table S10: Which are the bold values referred to in the table's description?

*** Apologies, this was missed before. We have bolded the relevant values.

We sincerely thank the editors at Communications Earth & Environment for their thorough and constructive feedback on our submitted manuscript, and for the opportunity to resubmit our revised work. We are also grateful to the reviewers for their valuable comments in the previous round of reviews. With three heat waves to date within this summer in the UK (temperatures peaking at 34°C) and the number of wildfire Fire and Rescue callouts in England and Wales double that of the extreme conditions of 2022 examined in this paper, these research findings are of ever increasing importance (we have referenced the most recent heatwaves in the introduction, line 50: “Three heat-wave events have occurred in the UK within 2025 alone and Europe has experienced four record-breaking heatwaves since 2003”). We have carefully addressed each point, with our responses provided in blue directly beneath each reviewer comment. Revisions in the manuscript are highlighted in yellow for ease of reference. Line numbers refer to those in the new version of the manuscript.

Most notably, within the revised manuscript we have directly addressed the concerns of the reviewers that all variables were previously equal, when some variables (e.g. weather) varied over time and some (landscape) were consistent over time within each site. We have therefore revised our statistical analysis. This revised analysis provides a more robust approach, supporting the results and conclusions of the previous manuscript. In this revised approach, several variables have been modified. Air temperature and relative humidity have been replaced with VPD on the suggestion of Reviewer 1. To simplify the analysis, soil classifications have now been grouped into five individual classes (Line 271: ‘We grouped the Soilscape categories into five groups to aid analysis, using the Soilscape descriptions: loamy & freely draining; sandy & freely draining; loamy & naturally wet; sandy & naturally wet; peaty & naturally wet (Supplementary Table 6).’). Further, to represent the control of climate and to reduce correlation between variables, the mean daily temperature has been replaced with the 20-year average of mean daily temperature for the day of sampling. (Line 262: ‘For each sample site, we calculated the long-term mean daily temperature of the site’s associated grid cell for the 20-year period 2004-2023 to represent long-term climatology of the site and time of year.’) This variable now directly represents the potential influence on FMC of the seasonal change in temperature and the long-term climatological differences across sites. Further, in line with the reviewer’s recommendations, we have removed the category of ‘temporal’ and replaced it with ‘phenological’, which includes NDVI and 20-year long term mean daily temperature. Sample year and time of day are now included merely as control variables and are not officially categorised in the results.

For our new analysis, we conducted a two-stage modelling approach in order to determine the effect of two types of variables on FMC; firstly, weather and phenological variables, which vary over time, and secondly, landscape variables which are consistent in sites across the entire timeframe. In the first stage, we created linear mixed models comparing FMC against weather and phenological variables with site as a random effect. We then took the effect sizes of sites and created a second model comparing these effects against the landscape variables associated with each site. This method firstly determined how much variation in FMC was related to weather, phenology and ‘Site’. Subsequently, the influence of each landscape variable on Site FMC was determined. The methods section has been fully updated to reflect the revised analysis approach and we have reworded the results to reflect the updated models. Within the revised methods we now state:

Line 269: “Using air temperature and relative humidity, we then calculated Vapour Pressure Deficit (VPD).”

Line 273: "For each sample site, we calculated the long-term mean daily temperature of the site's associated grid cell for the 20-year period 2004-2023 to represent long-term climatology of the site and time of year."

Line 281: "We grouped the Soilscape categories into five groups to aid analysis, using the Soilscape descriptions: loamy & freely draining; sandy & freely draining; loamy & naturally wet; sandy & naturally wet; peaty & naturally wet (Supplementary Table 6)."

Line 289: "We used a two-stage modelling approach for each of the three fuel types separately to determine the effects of weather, phenology and landscape variables on FMC. We chose a two stage modelling approach due to the temporal mismatch between (temporally varying) weather and phenology variables, and the (static) landscape variables. Firstly, we ran a mixed-effects linear model for FMC with fixed effects comprising weather variables (relative humidity, air temperature, number of days since rain and the total rainfall of the sampling day and five days prior (hereon called five-day rainfall)) and phenological variables (NDVI and long-term mean daily temperature), and with site as a random effect. Sample year and time of day were included as fixed effects to account for diurnal and interannual FMC variability. We extracted the random effect for each site and created a second linear model for these effect sizes with landscape variables (soil type, land cover type, elevation, aspect and slope) as fixed effects. Finally, we extracted the adjusted R-squared value from this landscape model. This method allowed us to investigate the effects of weather, phenology and landscape while accounting for the fact that weather and phenological variables vary over time, while landscape variables were consistent for each site. See Supplementary Tables 7-9 for model summaries and Supplementary Figs. 2 & 3 for landscape effect plots. The correlation between variables is shown in Supplementary Table 10. We ran all statistical analyses in R version 4.3.0⁵⁷."

And within the main section of the text:

Line 107: "These results present a two-stage modelling approach. The first model (mixed effects model) estimates the effect of weather and phenological variables on fuel moisture content (FMC), using sample site as a random effect. The second model (linear model) takes as a response variable the site level random effect to estimate the effect of landscape variables on FMC."

Line 112: "Dead heather FMC was significantly associated with weather variables only (VPD and total rainfall of the sampling day and five days prior (hereon called five-day rainfall); $p < 0.01$ for both) (Fig. 3a). The model showed no significant influence of phenology on dead heather FMC. Weather and phenological variables explained 47% of dead heather FMC variability, but the site-level random effect explained 0% of FMC variability indicating that landscape plays no significant role in dead heather FMC. Due to this lack of site-level variability, we were unable to run the landscape model for this fuel type. For live heather, in addition to the impact of VPD, FMC was also associated with phenological variables more than both the dead heather and organic layer; both long-term mean daily temperature and NDVI were significantly associated with FMC ($p < 0.01$ & $p = 0.02$ respectively) (Fig. 3b). Weather and phenological variables explained 23% of live heather variability, with a further 9% explained by site-level effects. Of these site-level effects, 37% of FMC variability was explained by landscape variables; peaty & naturally wet soil alone was significant (Fig. 3c; $p = 0.04$). The organic layer was significantly associated with a combination of weather variables and one phenological variable (long-term mean daily temperature) (Fig. 3d), but only 13% of FMC variability was explained by weather and phenological variables. 56% of FMC variability was explained by site-level effects, and of this 61% of site-level variability was explained by landscape variables; both soil type and elevation were significant (Fig. 3e; $p < 0.05$). Organic layer FMC was therefore influenced by landscape to a much greater degree than the other fuel types."

Reviewer #1 (Remarks to the Author):

Responses:

Thank you for this revision and for responding the comments of myself and the other reviewers. I appreciate the main point of the paper; that climate change is making regions more fire prone by removing some of the barriers to fire behaviour (even in locations that have not experience as much fire in the past). I think that this is a useful point to make, however I would like to see more explanation about how this is expected to be different from drivers of fire behaviour in locations where this is more understood (e.g., can you place this in the context of similar

research that has occurred in other, more fire-prone locations; and how would we expect fire in the UK to differ, or be similar to, other locations).

Within the original manuscript we highlighted the differences in controls between traditionally fire prone regions and within temperate regions. We stated:

Line 59: "The drying response of fuels to extreme weather in traditionally fire-prone Mediterranean and continental climates is established^{10,11}. However, we lack understanding of fuel moisture dynamics in non-fire-prone temperate regions."

And further we state:

Line 76: "Live *Calluna* provides the dominant fuel for fire spread¹⁹, the moisture of which exhibits complex dynamics²⁰, with ecohydrological drivers differing from weather dependent controls of dead fuels that underpin fuel moisture models in traditionally fire prone regions (e.g. Van Wagner & Pickett²¹). Notably, phenology and landscape provide strong impacts on the fuel moisture content of live vegetation^{22,23}. The moisture of live *Calluna* is lowest in the spring before the summer green up^{20,24}. As a result, large wildfires occur more frequently during spring rather than in warmer, drier summer conditions²⁵."

However, we recognise that we did not define the dominant controls on fuel moisture and associated fire danger indices within the traditionally fire prone region. Within the revised manuscript we now highlight the importance of fire weather in these fire prone regions and linked to associated literature. We therefore now further state:

Line 60: "Within these climates, fuel moisture and associated fire danger is determined predominantly by fire weather^{12,13}."

I also appreciate the attention paid to the revised statistical methods, although I have comments regarding that below. However, I still have concerns about the interpretations drawn from the data and the confusing and, in my opinion, somewhat arbitrary, categorization schema. Additional comments follow below:

We respond to these comments below where they are presented in further detail.

Categorization of variables. I think the categorization of variables is important because it has bearing on the conclusions drawn from Figure 3, one of the central points of the paper. 102-106. I still find it very confusing how temperature is in two different categories: I am not convinced that mean air temperature is a "temporal" variable while air temperature from a single point in time is a "weather" variable. Reviewer 2 also asked about this confusing separation, and I do not feel like a convincing explanation was given.

We recognise that the categorization, whilst important as you state, was confusing within the previous version of the manuscript. To enhance the clarity of the message being communicated we have changed the categorisation within the revised manuscript. We now have 1) Weather variables, specific to the day of sampling, 2) Phenological variables, which consists of NDVI and the long-term mean daily temperature averaged over 20 years (previously represented by the mean daily temperature), 3) Landscape variables, that represent the different characteristics of the site (i.e. soil type, aspect). Temporal variables (time of day and sample year) are ungrouped and are not presented in the results, as they were only included to account for diurnal and interannual controls on FMC that are not the focus of the manuscript.

112: "Mean air temperature and NDVI both represent the vegetation green up through the seasonal cycle". Day of the year would be a much more straightforward representation of vegetation green up than mean air temp. 115: "we acknowledge that mean temperature of sample day and NDVI therefore represent the phenological stage of the plant" – I am not

convinced that mean temperature of the sample day represents the phenological stage of the plant; day of the year would seem to be much more linked.

We recognise the complexity of the previous classification and have removed the use of the mean air temperature in accordance with the earlier comments. We instead apply the 20 year mean daily temperature. The 20-year average represents the climate and thus the phenological stage of the plants at the given location. It is a better representation than day of year, as it accounts for spatial variations in climate across the UK wide sampling design.

Language of causation

105: “Temporal variables provide an important control on the live *Calluna* canopy moisture content”: NDVI isn’t providing a “control” on moisture content (but other way around). And the other “temporal” variable of mean temperature (2nd most influential variable should still be a weather variable).

Use language of “association”/“correlation” not “control” or “driver”. You provide some tempering of this language in 115-116, but need to make sure this language is used throughout. For example in line 86: “We determined the primary controls on the fuel moisture” –NDVI is not a control on fuel moisture, but rather the other way around.

NDVI is an indicator of the phenology. The causal relationships between NDVI and fuel moisture content is difficult to define. Therefore, within the research question and the results section of the paper we have modified the language in line with Reviewer 1’s suggestions. This is to recognise that we are exploring the relationships between FMC and NDVI. Notably:

Line 90: we have changed this to ‘We determined the relationship between the fuel moisture of live *Calluna* canopy, dead *Calluna* canopy and organic ground fuel, and a variety of weather, phenological and landscape variables’

As the modelling approach has changed, we describe the results as FMC being ‘influenced by’ or ‘associated with’ different variables depending on the variable in question, e.g. ‘both long-term mean daily temperature and NDVI were significantly associated with FMC’ (line 119). Further, with the revised analysis approach, we no longer discuss the relationship between FMC and temporal variables. More broadly within the manuscript, when we are discussing more generally the weather, phenology, and landscape, these are direct controls on the fuel moisture and we have reviewed and retained this terminology where it is appropriate.

Model concerns

I would like to see how much variables are correlated! I’m sure lots of these variables are highly correlated. This makes it even harder to get at causal relationships as potentially confounding variables and reduce the relative influence of a variable, but that doesn’t mean it isn’t important mechanistically.

We have calculated the correlation between each variable within the weather/phenology model and have included them in supplementary material (Table S10). Because we have removed air temperature and replaced it (and relative humidity) with VPD, the highest correlation is now seen between NDVI and long-term mean daily temperature (0.35-0.46 across fuel types).

Why not use water balance terms (like VPD as Reviewer 2 recommends); you say it gets rid of the influence of temperature and humidity; but that would seem to be partly the point, in that

VPD is a more mechanistic/integrative metric of how plants experience temp/precip. Other papers that you cite (e.g., citation 9) also use VPD

126: Did models with VPD in them perform better? I would like to see justification for why they were not presented, not just that they replaced humidity or temp. In many ways (like the other reviewers pointed out), VPD would be a better, more mechanistic predictor so I would like to see justification for how models using VPD compare to models using temp & humidity.

We thank the reviewer for their suggestion and we appreciate the benefits of using VPD in the analysis, being more mechanistic and representative of the conditions which the plants experience. We have replaced air temperature and relative humidity with VPD in the first weather/phenology models. This alteration does not modify the research findings.

155-165 analysis: why are other variables not included in these models, like they are in the BRT models? It seems like you already modeled this earlier, so I'm curious why there is a need for two different modeling approaches instead of just using one to accomplish both goals (1. showing relative importance or effect size of predictors; 2. Showing model predictions). Even the comparison between years could be incorporated into the full model with an interaction between year and any relevant model; this would also allow for model selection to test whether an interaction with year improved model fit).

Within the revised manuscript we have removed these analyses – we agree with the reviewer that this further analysis is not required. The combination of the initial models describing the relationships between each variable and FMC, and the boxplots showing FMC across different time periods including the heatwave, are sufficient to capture the effect of the heatwave on FMC.

Different drivers of live vs. dead

It makes sense that live fuel moisture changes much more slowly than dead fuels, and has larger patterns of phenological changes. In other words, the scale of FMC change is different between live vs. dead. However, I don't think that Figure 3 provides convincing evidence that the two are being "driven" by wildly different things. If you look at what variables are in the top-5 highest relative influence between live and dead you get essentially the same variables for live and dead (not in order).

Live: Humidity; Temp (mean); Time of day (likely to be heavily correlated with temperature); Rainfall; NDVI

Dead: Humidity; Temp (mean); Temp (point in time); Rainfall; NDVI

Basically, it seems like the main argument (that live and dead fuel are driven by different things) relies on the definitions of categories of weather vs. temporal, but these categories are arbitrary and confusing, and ultimately, both live and dead have virtually the same variables as what is the most important.

Our new modelling approach generally gives the same clear message as our original BRT method. Dead heather FMC was significantly associated with weather variables only. In comparison, for live heather, in addition to the impact of VPD, FMC was also associated with phenological variables and site. For the organic layer, whilst FMC was significantly associated with a combination of weather variables and one phenological variable, 57% of FMC variability

was explained by site-level effects. However, in light of these comments and that above, we have modified our language throughout with regards to controls and drivers. For example, in the abstract, we now write (line 27): 'We show that the fuel moisture content of live fuel is associated significantly with phenological variables, dead fuel only with weather variables, whilst organic-rich ground fuels are associated with landscape variables more than other fuels constituents.'

Other comments:

72: "with ecohydrological drivers differing from weather dependent controls of dead fuels that underpin fuel moisture models in traditionally fire prone regions". How do they differ here compared to other locations that are more studied? There is also a phenological / seasonal pattern in more-studied locations as well.

We have added a sentence to the following section on line 59: "The drying response of fuels to extreme weather in traditionally fire-prone Mediterranean and continental climates is established^{10,11}. Within these climates, fuel moisture and associated fire danger is determined predominantly by fire weather^{12,13}."

155: missing "and" between "temperature" and "relative humidity"

We have replaced temperature and relative humidity with VPD so this has been removed.

Reviewer #2 (Remarks to the Author):

The authors have done a good job of addressing my previous comments. The addition of Figures 1 and 2 is particularly helpful, and the revised analysis using a GLM is a clear improvement in terms of statistical rigor. I think it is close, however I do feel there are several minor issues remaining.

In particular, I remain very unconvinced by the use of the term "temporal variable" and urge the authors to reconsider it:

- While you have added a definition of it in the Results (e.g., L 102), the term appears in the abstract where it is too vague to be meaningful.
- More importantly, the categorization itself is confusing. All weather variables are inherently temporal—they vary over time and reflect seasonal patterns—yet some are referred to as "weather variables" and others (e.g., mean temperature on the sampling day) are categorized as "temporal variables."
- NDVI does indeed vary temporally, but this is driven by weather and plant phenology, so why not just refer to it as a phenological variable, which can be easily understood.
- The variables grouped under the "temporal" label reflect very different processes occurring at different temporal scales. As such, I don't think this grouping adds clarity or analytical value.
- Moreover, there doesn't appear to be a unifying point or interpretation tied to the grouping of "temporal variables," so removing the label could simplify the paper and make the findings easier to interpret.

Instead, I suggest referring to each variable based on what it actually represents—for example:

- NDVI as a measure of plant phenology,
- Time of day as a sampling characteristic, or a “diel variable”
- Sample year as a statistical control.

Thank you for the thoughtful comments. Within the revised manuscript, and the revised analysis, we have closely followed the suggestions. We have changed our statistical methods (described fully at the start of this document). Part of this included changing the categorisation of NDVI and mean daily temperature (which we have changed from mean daily temperature of the day to a long-term mean temperature averaged over 20 years). These are now grouped as ‘phenological’, as suggested, and the other two temporal variables (time of day and sample year) are ungrouped and are not presented in the results, as they were only included to account for diurnal and interannual variation in FMC.

Additionally, I'm unsure why two separate variables are included to represent the temperature on the sampling day. I would expect these to be highly correlated.

Based on this comment and those of other reviewers, we have now removed mean daily temperature. This is replaced by the 20-year average temperature that represents the climate of the site more generally and can be used in conjunction with air temperature at the time of sampling.

On a related note, given the overlapping and potentially correlated nature of several predictors, did you test for correlations among all variables? This is important because of its implications on the relative importance results. You say that boosted regression trees can handle highly correlated predictors (L309-310), which I believe is really only true when prediction, as opposed to inference, is the goal. Calculating relative influence with two highly correlated variables in the same model will likely underestimate the importance of those variables (because omission of one of the variables will likely not cause significant loss of explanatory power while the other remains in the model). On this note, the method for calculating the relative influence scores is not clear to me (as these can be based on multiple methods), so please add explanation of that too. (Please correct me if I've missed this info).

We have changed the methods so no longer present relative influence scores in the text. We have calculated the correlation between each variable within the weather/phenology model and provide them in supplementary material (Table S10). Because we have removed air temperature and replaced it (and relative humidity) with VPD, the highest correlation is now seen between NDVI and long-term mean daily temperature (0.35-0.46 across fuel types).

Reviewer #3 (Remarks to the Author):

In my opinion, the authors have adequately addressed the points raised in the previous round. Explanations and clarifications have been incorporated to the text and new figures provide a more complete context for the study. Thanks to comments by other reviewers, the methods used for the analysis are now better justified. Therefore, I think the manuscript is now in better shape. I recommend it for publication, provided the following minor comments are considered and incorporated into the text:

“mean (air) temperature of the sample day”, regarded as a “temporal variable”, is interpreted

here as representative of the annual cycle of temperature (as I understand, this is indeed the intention with the “temporal” category of variables). However, it should not be confused/interchanged with “long-term mean temperature of a certain day”, which would be the average of mean temperature for a certain day over many years (e.g. 30, as traditionally used in climatology). The latter variable should be regarded as a temporal variable (though maybe not very informative/useful for the purposes of this study), especially considering that the “mean temperature of the sample day” might exhibit a considerable departure from it. Such anomaly, in contrast with the expected climatological value, could be in fact regarded as the consequence of “effective weather”. Then, shouldn’t be “mean (air) temperature of the sample day” considered a weather variable, along with “temperature at the time of sampling”? Please clarify and discuss.

We thank the reviewer for this comment and have now changed the mean daily temperature to a 20-year mean daily temperature. This should remove the possibility of anomalies, and also can represent regional differences in long-term climatologies across sample sites.

In line with previous comment, please revise lines 123 and 124: which variables represent seasonal and short-term temperature trends, respectively?

We have replaced mean daily temperature with long-term (20 year) mean daily temperature and replaced air temperature with VPD so this section has been rewritten.

Gamma GLM are now shown in Figure 5, considering July 2021 and July 2022 separately. I suggest fitting a third, more complete model (and plotting its corresponding curve) for each fuel type using the data from both periods. In my opinion, such models would represent a more general approach to the effective influence of weather (temperature) on the corresponding FMC.

We have removed these gamma GLMs based on comments by Reviewer 1. We do not feel that they added significantly to the message of the paper, and that the combination of the two-stage models and the comparison of FMC across regions sufficiently represent the effect of the heatwave on FMC.

*** We are very grateful to the editors and reviewers for their continued review of our manuscript, and we are delighted that Communications Earth & Environment would be happy to publish this manuscript with some small edits. We have addressed the final comments from Reviewer 3 below, marked by '***'.

-Modelling approach: please briefly introduce the “mixed effects model” following the author guidelines for this journal. Clarify whether there is a linear model implied and how the “estimates” corresponding to each variable (Fig. 3) should be interpreted.

*** We have added the following to the caption of Fig. 3: ‘Variables are significant if CI of estimate does not cross 0: negative estimate shows negative linear relationship between variable and FMC; positive estimate shows positive linear relationship between variable and FMC.’

-Definition and clarification of terms: in line with previous comment, particular attention should be paid to the results shown in Figure 3. The concept “effect size” for panels a,b,d just appears in this caption, and is referred to as “estimate” in the corresponding graphs. All terms used in panels and captions should be properly defined in the main text. Moreover and specifically, where is the exact definition of the “variable importance” quantified by t-values in Fig. 3c,e? Without any further methodological details, no clue is provided to the reader for the interpretation of these results.

Panels in Fig. 3c,e are even more delicate, since no indication is given to interpret the meaning of the labels. The caption even indicates “LC”. Again, all terms should be clearly defined in the main text and, in this case, also in Table 1.

*** We have changed ‘effect size’ to ‘estimate’ in the caption of Fig. 3 and have added the following to the methods on line 289: ‘Estimates calculated in the model depict the strength of the relationship (either positive or negative) of each variable with FMC variability (see Fig. 3).’

*** We have added the following to the methods on line 292: ‘We calculated the relative importance of each variable on across-site FMC variability by extracting each variable’s t-value, which represents the magnitude of the relationship between a predictor variable and the response variable. For categoric variables (soil type, land cover type), the model calculates a t-value for each category’s subgroup (e.g. for acid grassland, bog, coniferous forest and heathland within land cover). These were summed to give the overall importance for each categoric variable.’

*** We have added the following to the caption of Fig. 3: ‘Elev = elevation, LC = land cover, Soil = soil classification, Asp = aspect’

-Other comments:

Figure S2: Please describe in the caption all features for each panel. I am particularly curious on the fitted models for elevation, slope, and aspect. Same for similar figures.

*** We have edited the captions of Figure S2 and Figure S3 to the following: ‘Partial effect plots showing the relationship between across-site variation of FMC of *fuel type* and landscape factors associated with sample sites (a = land cover type; b = elevation (metres); c = slope (degrees); d = aspect (degrees); e = soil group (classified using Soilscapes (Farewell et al., 2011)). Model presented was the second model in a two-stage modelling process; the first mixed-effects linear model compared FMC with weather and phenological

variables, using sample site as a random effect. The effect sizes for sample site were extracted and used in a second linear model which compared these effect sizes to landscape variables.'